# Langevin Monte-Carlo Provably Learns Depth Two Neural Nets at Any Size and Data

## Abstract

In this note we prove that the Langevin Monte Carlo (LMC) algorithm can learn depth-2 neural nets of any size and for any data — and we give non-asymptotic convergence rates for it. We achieve this via obtaining distributional convergence of iterates for the stochastic training algorithm, LMC, on nets of any size. In particular, we show that in q-Rényi divergence, the iterates of LMC converge to the Gibbs distribution of certain Frobenius norm regularized losses for these nets, in both classification and regression settings. The amount of regularization needed for our results is independent of the size of the net. This work synthesizes several recent observations about isoperimetry conditions under which LMC converges and that two-layer neural loss functions can always be regularized by a certain constant amount such that they satisfy the Villani conditions, and thus their Gibbs measures satisfy a Poincaré inequality.

## 1  Introduction

Modern developments in artificial intelligence have significantly been driven by the rise of deep-learning. The highly innovative engineers who have ushered in this A.I. revolution have developed a vast array of heuristics that work to get the neural net to perform "human like" tasks. Most such successes can mathematically be seen to be solving the function optimization/"risk minimization" question, $\inf_{f \in \mathcal{F}} \mathbb{E}_{\boldsymbol{z} \in \mathcal{D}}[\ell(f, \boldsymbol{z})]$ where members of $\mathcal{F}$ are continuous functions representable by neural nets and $\ell : \mathcal{F} \times \mathrm{Support}(\mathcal{D}) \to [0, \infty)$ is called a "loss function" and the algorithm only has sample access to the distribution $\mathcal{D}$. Successful neural experiments can be seen as suggesting that there are many available choices of $\ell$, $\mathcal{F}$ & $\mathcal{D}$ for which highly accurate solutions to this seemingly extremely difficult question can easily be found. This is a profound mathematical mystery of our times.

The deep-learning technique that we focus on can be informally described as adding Gaussian noise to gradient descent. Works like Neelakantan et al. (2015) were among the earliest attempts to formally study how noisy gradient descent can outperform vanilla gradient descent for deep nets. In this work, we demonstrate how certain recent results can be carefully put together such that it leads to a first-of-its-kind development of our understanding of this ubiquitous method of training nets in realistic regimes of neural net training — hitherto unexplored by any other proof technique.

In Neelakantan et al. (2015) the variance of the noise was made step-dependent. However if the noise level is kept constant then this type of noisy gradient descent is what gets formally called as the Langevin Monte Carlo (LMC), also known as the Unadjusted Langevin Algorithm (ULA). For a fixed step-size $h > 0$ and an at least once differentiable "potential" function $V$, LMC can be defined by the following stochastic process in the domain of $V$ consisting of the parameter vectors $\boldsymbol{W}$,

$$\boldsymbol{W}_{(k+1)h} = \boldsymbol{W}_{kh} - h\nabla V(\boldsymbol{W}_{kh}) + \sqrt{2}(\boldsymbol{B}_{(k+1)h} - \boldsymbol{B}_{kh}) \tag{1}$$

Here, the Brownian increment $\boldsymbol{B}_{(k+1)h} - \boldsymbol{B}_{kh}$ follows a normal distribution with mean 0 and variance $h$. Thus, if one has oracle access to the gradient of the potential $V$ and the ability to sample Gaussian random variables then it is straightforward to implement this algorithm. This $V$ can be instantiated as the objective of an optimization problem, such as the empirical loss function in a machine learning setup on a class of predictors parameterized by the weight $\boldsymbol{W}$. Then this approach is analogous to perturbed gradient descent, for which a series of recent studies have provided proofs demonstrating its effectiveness in escaping saddle

points Jin et al. (2021). Intuition suggests that LMC would asymptotically sample from the Gibbs measure $\exp(-V)$ and thus achieve learning. However, proving this has been challenging, and in the later sections, we will review the progress made so far – and we will tie them to formalize this "folklore" specific to neural nets.

## 1.1 Summary of Results

We consider the standard empirical losses for depth-2 nets of arbitrary width while using arbitrary data and initialization of weights, in both regression as well as classification setups — while the loss is regularized by a certain constant amount. In Theorem 4.1, we establish that LMC on this empirical risk can minimize the corresponding neural population risk, thereby *we achieve a first-of-its-kind provable learning of neural nets for any size.*

Towards proving Theorem 4.1, we establish in Theorem 4.2 the critical result that the iterates of the LMC algorithm exhibit $\tilde{O}(\varepsilon)$ close distributional convergence in $q$-Rényi divergence to the Gibbs measure of the empirical loss at a rate of $\tilde{O}\left(\frac{1}{\varepsilon}\right)$. It is to be noted that the convergence rates scale with the Poincaré constant of the induced Gibbs' measure and avoids using the Log-Sobolev inequality (which would be expected to lead to larger isoperimetry constants) and this requires certain technical workarounds to be developed in the intermediate lemmas (given in Appendix B.2) that pertain to the stability of the "Gibbs' Algorithm" that are needed to prove Theorem 4.1.

We further note that the threshold amount of regularization needed in the above is *independent of the width of the nets.* Further, this threshold value is proportionately small if the norms of the training data are small or the threshold value can be made arbitrarily small by choosing outer layer weights to be similarly small.

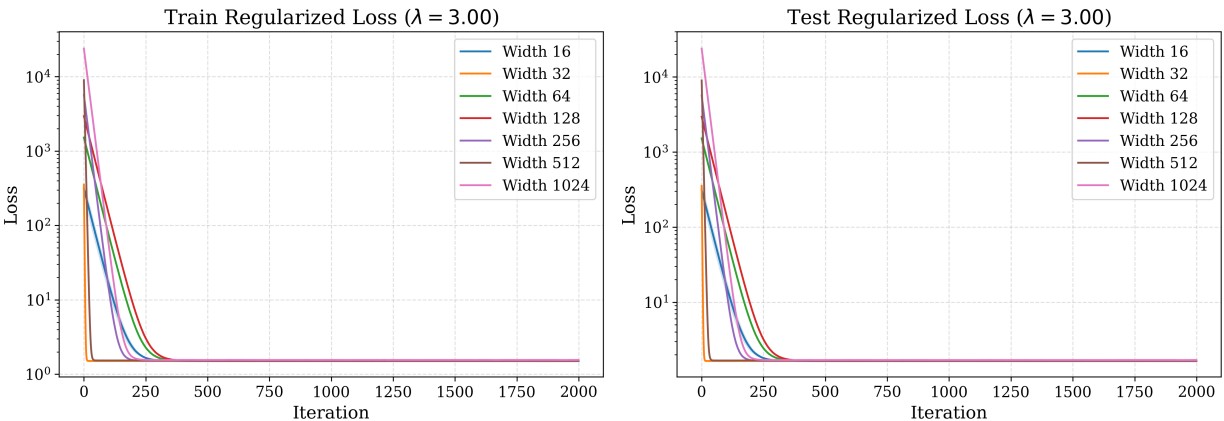

Figure 1: In this figure, we show the train and test error convergence plots for depth-2 neural networks at varying widths when trained via LMC for a regression task.

Figure 1 illustrates the phenomena captured by Theorem 4.1. We consider a regression task and use LMC to train depth-2 tanh activated networks at varying widths, while keeping the norm of the second layer fixed to ensure that each setup has the same critical value of weight regularization parameter ($\lambda$) at which Theorem 4.1 begins to apply for it. For each setup $\lambda$ is set above this common critical value, and we can observe that it induces simultaneous training and learning to happen for all the widths. Further, we note that the convergence behaviours differ with widths indicating that the dynamics of LMC is not dictated by the regularization and hence evidencing that the regularization is not very strong. In Section 5 Figure 2, we provide further details on the setup of this experiment, along with a demonstration that when training with noisy data, the regularized loss trains to worse values (differently for train and test) as the label noise increases — and that this sensitivity persists at different widths — thus further evidencing that the required regularization for the proofs here, is not the determinant of the LMC dynamics. Lastly, in Section 5.1, we present the corresponding experiments using the AdamW optimizer Loshchilov & Hutter (2019) and observe comparable performance and hence demonstrating that the setup used here for theory can mimic the heuristics popularly deployed for learning neural nets.

## 1.2 Comparison To Existing Literature

In the forthcoming section we will attempt an overview of the state-of-the-art results for both the ideas involved here, that of provable deep-learning and provable convergence of Langevin Monte-Carlo. Here we summarize the salient features that make our Theorem 4.1 a distinct improvement over existing results.

*Firstly,* we note that to the best of our knowledge, there has never been a convergence result for the law of the iterates of any stochastic training algorithm for neural nets. And that is amongst what we achieve in our key results.

In the last few years, there has been a surge in the literature on provable training of various kinds of neural nets. But the convergence guarantees in the existing literature either require some minimum neural net width – growing w.r.t. inverse accuracy and the training set size (NTK regime Chizat et al. (2019); Du et al. (2018)), infinite width (Mean Field regime Chizat & Bach (2018); Chizat (2022); Mei et al. (2018)) or other assumptions on the data when the width is parametric, like assumptions on the data labels being produced by a net of the same architecture as being trained Ge et al. (2019); Zhou et al. (2021).

We note that unlike as in the mean-field regime results Chizat (2022); Nitanda et al. (2022); Nitanda (2024), where distributional convergence results needed the unrealistically strong infinite-width assumptions we do not make assumptions on either data or the size of the net, while obtaining similar distributional convergence.

*Secondly,* as reviewed in Section 2.1 we recall that convergences of algorithms for training neural nets have always used special initializations and particularly so when the width of the net is unconstrained.

In comparison, we note that our convergence result is parametric in initialization and hence allows for a wide class of initial distributions on the weights of the net. This flexibility naturally exists in the recent theorems on convergence of LMC and we inherit that advantage because of being able to identify the neural training scenarios that fall in the ambit of these results.

*Thirdly,* — and importantly — we posit that the methods we outline for proving LMC convergence for realistic neural net losses are highly likely to be adaptable to more complex machine learning scenarios than considered here. Certain alternative methods (as outlined in the conclusion in Section 6) of proving isoperimetric inequalities for log-concave distribution are applicable to the cases we consider but they exploit special structures which are not as amenable to more complex scenarios as proving the loss function to be of the Villani type, as is the method here. Thus, a key contribution of this work is to demonstrate that this Villani-function based proof technique is doable for realistic ML scenarios and hence we open multiple avenues of future research.

## 2 Related Works

There is a vast literature on provable deep-learning and Langevin Monte-Carlo algorithms and giving a thorough review of both the themes is beyond the scope of this work. In the following two subsections we shall restrict ourselves to higlighting some of the papers in these subjects respectively, and we lean towards the more recent results.

## 2.1 Review of Works on Provable Deep-Learning

One of the most popular regimes for theory of provable training of nets has been the so-called "NTK" (Neural Tangent Kernel) regime – where the width is a high degree polynomial in the training set size and inverse accuracy (a somewhat *unrealistic* setup) and the net's last layer weights are scaled inversely with square-root of the width, Du et al. (2018); Su & Yang (2019); Allen-Zhu et al. (2019b); Du & Lee (2018); Allen-Zhu et al. (2019a); Arora et al. (2019b); Li et al. (2019); Arora et al. (2019a); Chizat et al. (2019). The core insight in this line of work can be summarized as follows: for large enough width and scaling of the last layer's weights as given above, (Stochastic) Gradient Descent *with certain initializations* converges to a function that fits the data perfectly, with minimum norm in a Reproducing Kernel Hilbert Space (RKHS) defined by the neural tangent kernel – that gets specified entirely by the initialization A key feature of this regime is that the net's matrices do not travel outside a constant radius ball around the starting point – a property that is often not true for realistic neural training scenarios. To overcome this limitation of NTK, Chizat et al. (2019); Chizat (2022); Mei et al. (2018) showed that training is also provable in a different asymptotically large

width regime, the mean-field, which needs an inverse width scaling of the outer layer — as opposed to the inverse square-root width scaling for the same that induces the NTK regime. In the mean-field regime of training, the parameters are not confined near their initialization and thereby allowing the model to explore a richer class of functions.

In particular, for the case of depth 2 nets – with similarly smooth gates as we focus on – and *while not using any regularization*, in Song et al. (2021) global convergence of gradient descent was shown using number of gates scaling sub-quadratically in the number of data. On the other hand, for the special case of training depth 2 nets with ReLU gates on cross-entropy loss for doing binary classification, in Ji & Telgarsky (2020) it was shown that one needs to blow up the width poly-logarithmically with inverse target accuracy to get global convergence for SGD. But compared to NTK results cited earlier, in Ji & Telgarsky (2020) the convergence speed slows down to a polynomial in the inverse target accuracy.

### 2.1.1 Need And Attempts To Go Beyond Large Width Limits of Nets

The essential proximity of the NTK regime to kernel methods and it being less powerful than finite nets has been established from multiple points of view. Allen-Zhu & Li (2019); Wei et al. (2019).

Specific to depth-2 nets — as we consider here — there is a stream of literature where analytical methods have been honed to this setup to get good convergence results without width restrictions, while making other structural assumptions about the data or the net. Janzamin et al. (2015) was one of the earliest breakthroughs in this direction and for the restricted setting of realizable labels they could provably get arbitrarily close to the global minima. For non-realizable labels they could achieve the same while assuming a large width but in all cases they needed access to the score function of the data distribution which is a computationally hard quantity to know. More recently, Awasthi et al. (2021) have improved the above paradigm to include ReLU gates while being restricted to the setup of realizable data and its marginal distribution being Gaussian.

One of the first proofs of gradient based algorithms doing neural training for depth−2 nets appeared in Zhong et al. (2017). In Ge et al. (2019) convergence was proven for training depth-2 ReLU nets for data being sampled from a symmetric distribution and the training labels being generated using a 'ground truth' neural net of the same architecture as being trained — the so-called "Teacher–Student" setup. For similar distributional setups, Karmakar et al. (2023) identified classes of depth-2 ReLU nets where they could prove linear-time convergence of training — and they also gave guarantees in the presence of a label poisoning attack. The authors in Zhou et al. (2021) consider a different Teacher–Student setup of training depth 2 nets with absolute value activations, where they can get convergence in polynomial time, under the restrictions of assuming Gaussian data, initial loss being small enough, and the teacher neurons being norm bounded and 'well-separated' (in angle magnitude). Cheridito et al. (2022) get width independent convergence bounds for Gradient Descent (GD) with ReLU nets, however at the significant cost of restricting to only an asymptotic guarantee and assuming an affine target function and one–dimensional input data. While being restricted to the Gaussian data and the realizable setting for the labels, an intriguing result in Chen et al. (2021) showed that fully poly-time learning of arbitrary depth 2 ReLU nets is possible if one can adaptively choose the training points, the so-called "black-box query model".

### 2.1.2 Provable Training of Neural Networks Using Regularization

Using a regularizer is quite common in deep-learning practice and recently a number of works have appeared which have established some of these benefits rigorously. In particular, Wei et al. (2019) showed a specific classification task (noisy–XOR) definable in any dimension $d$ s.t no 2 layer neural net in the NTK regime can succeed in learning the distribution with low generalization error in $o(d^2)$ samples, while in $O(d)$ samples one can train the neural net using Frobenius/$\ell_2$−norm regularization.

### 2.2 Review of Provable Distributional Convergence of Langevin Monte-Carlo Algorithm

There has been a flurry of activity in recent times to derive non-asymptotic distributional convergence of LMC entirely from assumptions of smoothness of the potential and the corresponding Gibbs measure satisfying functional inequalities. In Dalalyan (2017) it was proved that LMC converges in the Wasserstein metric ($W_2$) for potentials that are strongly convex and gradient-Lipschitz. The key idea that lets proofs go

beyond this and have LMC convergence happen for non-convex potentials $V$ is to be able to exploit the fact that corresponding Gibbs measure ($\sim e^{-V}$) might satisfy certain isoperimetric/functional inequalities.

Two of the functional inequalities that we will often refer to in this section are the Poincaré inequality (PI) and the log-Sobolev inequality (LSI). A distribution $\pi$ is said to satisfy the PI for some constant $C_{PI}$, if for all smooth functions $f : \mathbb{R}^d \to \mathbb{R}$,

$$\mathrm{Var}_\pi(f) \le C_{PI}\mathbb{E}_\pi[\|\nabla f\|^2] \tag{2}$$

Similarly, we say that $\pi$ satisfies an LSI for some constant $C_{LSI}$, if for all smooth $f : \mathbb{R}^d \to \mathbb{R}$, $\mathrm{Ent}_\pi(f^2) \le 2C_{LSI}\mathbb{E}_\pi[\|\nabla f\|^2]$, where $\mathrm{Ent}_\pi(f^2) := \mathbb{E}_\pi[f^2 \ln\left(\frac{f^2}{\mathbb{E}_\pi(f^2)}\right)]$.

The Poincaré inequality is strongly motivated as a relevant condition to be satisfied by a measure because of its relation to ergodicity. A defining property of it is that a Markov semigroup exhibits exponentially fast mixing to its stationary measure, in the $L_2$ metric, iff the stationary measure satisfies the Poincaré inequality Handel (2016). Assuming LSI on the stationary measure would further ensure exponentially fast mixing of their relative entropy distance Bakry et al. (2014).

In the landmark paper Raginsky et al. (2017), it was pointed out that one can add a regularization to a potential and make it satisfy the dissipativity condition so that Stochastic Gradient Langevin Dynamics (SGLD) provably converges to its global minima. We recall that a function $f$ is said to be $(m, b)$–dissipative, if for some $m > 0$ and $b \ge 0$ we have, $\langle \boldsymbol{x}, \nabla f(\boldsymbol{x})\rangle \ge m\|\boldsymbol{x}\|^2 - b \quad \forall \boldsymbol{x} \in \mathbb{R}^d$

The key role of the dissipativity assumption was to lead to the LSI inequality to be valid. We note that subsequently considerable work has been done where the convergence analysis of Langevin dynamics is obtained with dissipativity being assumed on the potential Erdogdu et al. (2018); Erdogdu & Hosseinzadeh (2021); Erdogdu et al. (2022); Mou et al. (2022); Nguyen et al. (2023).

In a significant development, in Vempala & Wibisono (2019) it was shown that if isoperimetry assumptions such as Poincaré or Log-Sobolev inequality are made (without explicit need for dissipativity) on appropriate measures derived from a smooth potential, then it's possible to prove convergence of LMC in the $q$–Rényi metric, for $q \ge 2$. This is particularly interesting because it follows that under PI, a convergence in 2-Rényi would also imply a convergence in the total variation, the Wasserstein distance and the KL divergence Liu (2020).

It is also notable, that unlike previous works Jordan et al. (1998); Jiang (2021), in Vempala & Wibisono (2019) only the Lipschitz smoothness of the gradient is needed and smoothness of higher order derivatives is not required, once functional inequalities get assumed for the mixing measure. But, the only case in Vempala & Wibisono (2019) where convergence (in KL) is shown via assumptions being made solely on the potential, LSI is assumed on the corresponding Gibbs measure. While Vempala & Wibisono (2019) also proved LMC convergence while assuming PI, which is weaker than LSI, the assumption is made on the mixing measure of the LMC itself – an assumption which is hard to verify a priori. Being able to bridge this critical gap, can be seen as one of the strong motivations that drove a sequence of future developments, which we review next.

We recall the Latała-Oleskiewicz inequality (LOI) Latała & Oleszkiewicz (2000). which is a functional inequality that interpolates between PI and LSI. We say $\pi$ satisfies the LOI of order $\alpha \in [1, 2]$ and constant $C_{LOI(\alpha)}$ if for all smooth $f : \mathbb{R}^d \to \mathbb{R}$, $\sup_{p\in(1,2)} \frac{\mathbb{E}_\pi(f^2) - \mathbb{E}_\pi(f^p)^{2/p}}{(2-p)^{2(1-1/\alpha)}} \le C_{LOI(\alpha)}\mathbb{E}_\pi[\|\nabla f\|^2]$. This inequality is equivalent to PI at $\alpha = 1$, and LSI at $\alpha = 2$. Convergence under this general isoperimetry condition and for general smoothness conditions on the potential have been established in Chewi et al. (2024).

Earlier, an intriguing result in Balasubramanian et al. (2022) showed that it is possible to obtain a convergence for the time averaged law of the LMC in Fisher Information to the Gibbs measure of the potential, without any isoperimetry assumptions on it. But in Proposition 1 of Balasubramanian et al. (2022), examples are provided of two sequences of measures that converge in the Fisher information metric but not in Total Variation.

Hence, it was further shown in Balasubramanian et al. (2022) that if Poincaré condition is assumed then this convergence can also be lifted to the TV metric. Under the same assumption of PI on the Gibbs measure,

in comparison to Chewi et al. (2024), here the rate of convergence is given for the averaged measure and it has better dependence on the dimension but worse dependence on the accuracy. But for any given target error the result in Balasubramanian et al. (2022) implies faster convergence when the dimension (in our case, the number of trainable parameters in the network) exceeds the inverse of the target error – which is not uncommon for neural networks. *In Section 4, we will revisit Chewi et al. (2024) in further details, as our key result would follow from being able to invoke a result contained therein.*

## 3 The Mathematical Setup of Neural Nets and Langevin Monte-Carlo

In this segment we will define the neural net architecture, the loss functions and the algorithm for which we will prove our learning guarantees.

**Definition 1 (The Depth-2 Neural Loss Functions).** Let, $\sigma : \mathbb{R} \to \mathbb{R}$ (applied element-wise for vector valued inputs) be at least once differentiable activation function. Corresponding to it, consider the width $p$, depth 2 neural nets with fixed outer layer weights $\boldsymbol{a} \in \mathbb{R}^p$ and trainable weights $\boldsymbol{W} \in \mathbb{R}^{p \times d}$ as, $\mathbb{R}^d \ni \boldsymbol{x} \mapsto f(\boldsymbol{x}; \boldsymbol{a}, \boldsymbol{W}) = \boldsymbol{a}^\top \sigma(\boldsymbol{W}\boldsymbol{x}) \in \mathbb{R}$ and the regularized loss function, for any $\lambda > 0$, is defined as, $\tilde{L}_{\mathcal{S}_n}(\boldsymbol{W}) \coloneqq \frac{1}{n}\sum_{i=1}^n \tilde{L}'_i(\boldsymbol{W}) + \frac{\lambda}{2}\|\boldsymbol{W}\|_F^2$, where $\mathcal{S}_n \in X^n$ is a set of $n$ training data points sampled i.i.d from $X = \mathbb{R}^d \times \mathbb{R}$ and $\tilde{L}'_i$ is the loss evaluated on the $i^{th}$ of them.

Then corresponding to a given set of $n$ training data $(\boldsymbol{x}_i, y_i) \in \mathbb{R}^d \times \mathbb{R}$, with $\|\boldsymbol{x}_i\|_2 \leq B_x, |y_i| \leq B_y, \ i = 1, \ldots, n$ the mean squared error (MSE) loss function for each data point is defined by $\tilde{L}'_i(\boldsymbol{W}) \coloneqq \frac{1}{2}\left(y_i - f(\boldsymbol{x}_i; \boldsymbol{a}, \boldsymbol{W})\right)^2$. Similarly, if we consider the set of $n$ binary class labeled training data $(\boldsymbol{x}_i, y_i) \in \mathbb{R}^d \times \{+1, -1\}$, with $\|\boldsymbol{x}_i\|_2 \leq B_x, \ i = 1, \ldots, n$ then we can define the binary cross entropy (BCE) loss for each data point by $\tilde{L}'_i(\boldsymbol{W}) \coloneqq \log\left(1 + e^{-y_i f(\boldsymbol{x}_i; \boldsymbol{a}, \boldsymbol{W})}\right)$.

**Definition 2 (Properties of the Activation $\sigma$).** Let the $\sigma$ used in Definition 1 be bounded s.t. $|\sigma(x)| \leq B_\sigma$, $C^\infty$, $L$–Lipschitz and $L'_\sigma$–smooth (gradient-Lipschitz). Further assume that there exist a constant vector $\boldsymbol{c}$ and positive constants $M_D$ and $M'_D$ s.t. $\sigma(\boldsymbol{0}) = \boldsymbol{c}$ and $\forall x \in \mathbb{R}, |\sigma'(x)| \leq M_D, |\sigma''(x)| \leq M'_D$.

We note that the standard sigmoid and the tanh gates satisfy the above conditions. Next, we shall formally define the necessary isoperimetry condition and recall in the lemmas immediately following it that this condition can be true for the Gibbs' measure of the neural losses defined above.

**Definition 3 (Poincaré-type Inequality (PI)).** A measure $\mu$ is said to satisfy the Poincaré-type inequality if $\exists\, C_{PI} > 0$ such that $\forall h \in C_c^\infty(\mathbb{R}^d)$, $\mathrm{Var}_\mu[h] \leq C_{PI} \cdot \mathbb{E}_\mu[\|\nabla h\|^2]$, where $C_c^\infty(\mathbb{R}^d)$ denotes the set of all compactly supported smooth functions from $\mathbb{R}^d$ to $\mathbb{R}$.

In terms of the above, we state the crucial intermediate lemmas quantifying the niceness of the empirical losses that we consider.

**Lemma 3.1 (Classification with Binary Cross Entropy Loss).** In the setup of binary classification as contained in Definition 1, and the given definition $M_D$ and $L$ as given in Definition 2 above, there exists a constant $\lambda_c^{\mathrm{BCE}} = \frac{M_D L B_x^2 \|a\|_2^2}{2}$ s.t $\forall \lambda > \lambda_c^{\mathrm{BCE}}$ and $s > 0$ the Gibbs measure $\sim \exp\left(-\frac{2\tilde{L}}{s}\right)$ satisfies a Poincaré-type inequality (Definition 3). Moreover, if the activation satisfies the conditions of Definition 2 then $\exists\, \beta_{\mathrm{BCE}} > 0$ s.t. the empirical loss, $\tilde{L}_{\mathcal{S}_n}$ is gradient-Lipschitz with constant $\beta_{\mathrm{BCE}}$, and ,

$$\beta_{\mathrm{BCE}} \leq \sqrt{p}\left(\frac{\sqrt{p}\|\boldsymbol{a}\|_2 M_D^2 B_x}{4} + \left(\frac{2 + \|c\|_2 + \|\boldsymbol{a}\|_2 B_\sigma}{4}\right)M'_D B_x p + \lambda\right) \tag{3}$$

**Lemma 3.2 (Regression with Squared Loss).** In the setup of Mean Squared Error as contained in Definition 1 and given the definition of $M_D$ and $L$ as given in Definition 2, there exists a constant $\lambda_c^{\mathrm{MSE}} \coloneqq 2 M_D L B_x^2 \|\boldsymbol{a}\|_2^2$ s.t $\forall\, \lambda > \lambda_c^{\mathrm{MSE}}$ & $s > 0$, the Gibbs measure $\sim \exp\left(-\frac{2\tilde{L}}{s}\right)$ satisfies a Poincaré-type inequality (Definition 3). Moreover, if the activation satisfies the conditions in Definition 2 then $\exists\, \beta_{\mathrm{MSE}} > 0$ s.t. the empirical loss, $\tilde{L}_{\mathcal{S}_n}$ is gradient-Lipschitz with constant $\beta_{\mathrm{MSE}}$, and,

$$\beta_{\mathrm{MSE}} \leq \sqrt{p}\left(\|\boldsymbol{a}\|_2 B_x B_y L'_\sigma + \sqrt{p}\|\boldsymbol{a}\|_2^2 M_D^2 B_x^2 + p\|\boldsymbol{a}\|_2^2 B_x^2 M'_D B_\sigma + \lambda\right) \tag{4}$$

The full proofs of the above two lemmas can be be found in Gopalani et al. (2024) and Gopalani & Mukherjee (2025) respectively.

### 3.1 Villani Functions

In the proofs of Lemmas 3.1 and 3.2 in Gopalani & Mukherjee (2025); Gopalani et al. (2024), the primary strategy for showing that the Gibbs measure of the loss functions of certain depth-2 neural networks satisfy the Poincaré inequality, involved first proving that these measures are Villani functions. Below, we define the criterion that characterize a Villani function.

**Definition 4** (**Villani Function**(Shi et al. (2023); Villani (2006))). A map $f : \mathbb{R}^d \to \mathbb{R}$ is called a Villani function if it satisfies the following conditions, **1.** $f \in C^\infty$, **2.** $\lim_{\|x\| \to \infty} f(x) = +\infty$, **3.** $\int_{\mathbb{R}^d} \exp\left(-\frac{2f(x)}{s}\right) dx < \infty \ \forall s > 0$, and **4.** $\lim_{\|x\| \to \infty} \left(-\Delta f(x) + \frac{1}{s} \cdot \|\nabla f(x)\|^2\right) = +\infty \ \forall s > 0$. Further, any $f$ that satisfies conditions 1 – 3 is said to be "confining".

The following lemma from Shi et al. (2023) can be invoked to determine that the Gibbs measure corresponding to Villani functions satisfies a Poincaré-type inequality.

**Theorem 3.3** (Lemma 5.4 in Shi et al. (2023)). Given $f : \mathbb{R}^d \to \mathbb{R}$, a Villani function (Definition 4), for any given $s > 0$, we define a measure with density, $\mu_s(x) = \frac{1}{Z_s} \exp\left\{-\frac{2f(x)}{s}\right\}$, where $Z_s$ is a normalization factor. Then this (normalized) Gibbs measure $\mu_s$ satisfies a Poincaré-type inequality (Definition 3) for some $C_{PI} > 0$ (determined by $f$).

### 3.2 The Langevin Monte Carlo Algorithm

The algorithm we study for the nets defined earlier in this section can be formally defined as follows.

**Definition 5** (**Langevin Monte Carlo Algorithm**). Denoting the step-size as $h > 0$, the Langevin Monte Carlo (LMC) algorithm, corresponding to an objective function $\frac{2\tilde{L}_{\mathcal{S}_n}}{s}$, where $\tilde{L}_{\mathcal{S}_n}$ is the loss function as defined in Definition 1 and $s > 0$ is an arbitrary constant, is defined as $W_{(k+1)h} = W_{kh} - \frac{2h}{s} \nabla \tilde{L}_{\mathcal{S}_n}(W_{kh}) + \sqrt{2}(B_{(k+1)h} - B_{kh})$.

Here, $(B_t)_{t \geq 0}$ is a standard $(p \times d)$-dimensional Brownian motion. We also need the continuous-time interpolation of the above LMC algorithm which is defined as,

**Definition 6** (**Continuous-Time Interpolation of LMC**). Using the setup of Definition 5, the continuous-time interpolation of the LMC is defined as $W_t \coloneqq W_{kh} - \frac{2(t-kh)}{s} \nabla \tilde{L}_{\mathcal{S}_n}(W_{kh}) + \sqrt{2}(B_t - B_{kh})$ for $t \in [kh, (k+1)h]$. We denote the law of $W_t$ as $\pi_t$.

## 4 Main Result on Langevin Monte Carlo Provably Learning Nets of Any Width and for Any Data

Given the formal setup in the previous section, we can state our key result on population risk minimization by LMC for neural losses considered in Lemmas 3.1 and 3.2. To this end, we recall that for two probability measures $\mu$ and $\pi$, Rényi divergence metric of order $q \in (1, \infty)$, is defined as $R_q(\mu \| \pi) \coloneqq \frac{1}{q-1} \ln \left\| \frac{d\mu}{d\pi} \right\|_{L_q(\pi)}^q$ and the 2-Wasserstein distance as $W_2(\mu \| \pi) \coloneqq \inf\{(\mathbb{E}[\|U - V\|_2^2])^{1/2} : U \sim \mu, V \sim \pi\}$.

**Theorem 4.1.** Consider the regularized empirical loss $\tilde{L}_{\mathcal{S}_n}(W)$ for neural networks, as defined in Definition 1, and recall that the corresponding Gibbs measure $\mu_s$ satisfies the Poincaré Inequality for some constant $C_{PI} > 0$ (Definition 3), when the loss is regularized as specified in Lemma 3.2 for the squared loss and Lemma 3.1 for the logistic loss. Then, there exist constants $\tilde{C}_3$ (which is linear in $C_{PI}$), $m, b, B > 0$ that depend on the loss function $\tilde{L}_{\mathcal{S}_n}(W)$, as well as a constant $\kappa_0 < \infty$ that exists for any "nice" initial distribution $\pi_0$, such that for any $s \leq \min(2, m)$, $\varepsilon > 0$, and for suitably chosen $N, h > 0$ (as given below) the expected excess risk of $W_{Nh}$ is bounded as,

$$\mathbb{E}[\mathcal{R}(W_{Nh})] - \mathcal{R}^* \leq \frac{\tilde{C}_3}{n} + \frac{pds}{4} \log\left(\frac{e\beta}{m}\left(\frac{2b}{spd} + 1\right)\right) + \left(\beta\sqrt{\kappa_0 + 2 \cdot \max\left(1, \frac{1}{m}\right)\left(b + 2B^2 + \frac{pds}{2}\right)} + B\right) 2C_{PI}\varepsilon$$

where, $n$ is the number of training samples for the loss function $\tilde{L}_{\mathcal{S}_n}(W)$, $\mathcal{R}(W) \coloneqq \mathbb{E}_{S_n}[\tilde{L}_{\mathcal{S}_n}(W)]$ and $\mathcal{R}^* \coloneqq \inf_{W \in \mathbb{R}^{p \times d}} \mathcal{R}(W)$.

**Remark.** $\beta$ is understood to be $\beta_{\text{MSE}}$ or $\beta_{\text{BCE}}$ as given in Lemma 3.2 for the squared loss and Lemma 3.1 for the logistic loss, as the case may be. Here, $m$ and $b$ are the "dissipativity" constants, $B$ denotes the upper bound on the gradient of the loss at $W = 0$, and the conditions for the "nice" initialization of the weights

are further detailed in the claims in Appendix B.1. $\tilde{C}_3$ depends on the properties of the loss function and the data — whose exact analytic expression is given in Appendix B.2.

The number of steps $N$ and the step-size $h$ needed for the above guarantee to hold is s.t $Nh = \tilde{\Theta}\left(C_{PI}R_3(\pi_0\|\mu_s)\right)$ and $h = \tilde{\Theta}\left(\frac{\ln(\varepsilon+1)}{pdC_{PI}\ \tilde{\beta}(L_0,\beta)^2\ R_3(\pi_0\|\mu_s)} \times \min\left\{1, \frac{1}{2\ln(\varepsilon+1)}, \frac{pd}{r}, \frac{pd}{R_2(\pi_0\|\mu_s)^{1/2}}\right\}\right)$, here $\tilde{\beta}(L_0,\beta)$ is a constant that depends on, $L_0 := \nabla\tilde{L}_{\mathcal{S}_n}(0)$ and $\beta$, where $\beta$ is the gradient-Lipschitz constant of the loss, and we define $r := \int \|\boldsymbol{W}\| d\mu_s$.

The proof of the above theorem is provided in Appendix B.3. At the end of Appendix B.3, we provide a heuristic argument demonstrating that the upper bound on the expected excess risk proven above can be made $\tilde{O}(\varepsilon)$ for any $\varepsilon > 0$ for large enough $n$ and $s = O(\epsilon)$ — a behavior which is also verified in the experiments in Section 5. And thus we can recover the non-asymptotic convergence rate that was mentioned in the summary in Section 1.1.

Towards obtaining the main result stated above we prove the following key theorem about distributional convergence of LMC on the neural nets that we consider.

**Theorem 4.2** (**Convergence of LMC in $q$-Rényi for Appropriately Regularized Neural Nets**). Continuing in the setup of the loss as required in Theorem 4.1, we recall the Gibbs measure corresponding to the LMC objective function as $\mu_s \propto \exp\left\{\frac{-2\tilde{L}_{\mathcal{S}_n}}{s}\right\}$.

We assume that $\varepsilon^{-1}, r, C_{PI}, \tilde{\beta}(L_0,\beta), R_2(\pi_0\|\mu_s) \geq 1$ and $q \geq 2$. Then, LMC with a step-size

$$h_q = \tilde{\Theta}\left(\frac{\varepsilon}{pdq^2C_{PI}\ \tilde{\beta}(L_0,\beta)^2\ R_{2q-1}(\pi_0\|\mu_s)} \times \min\left\{1, \frac{1}{q\varepsilon}, \frac{pd}{r}, \frac{pd}{R_2(\pi_0\|\hat{\mu}_s)^{1/2}}\right\}\right),$$

satisfies $R_q(\pi_T\|\mu_s) \leq \varepsilon$ where $\pi_T$ is the law of the iterate of the interpolated LMC (Definition 6), with $T = N_qh_q = \tilde{\Theta}\left(qC_{PI}R_{2q-1}(\pi_0\|\mu_s)\right)$. Here, $C_{\text{PI}}$ denotes the Poincaré constant corresponding to the Gibbs measure $\mu_s$ satisfying a Poincaré-type inequality and $\hat{\mu}_s \propto \exp\left(-\hat{V}\right)$ where, $\hat{V} := \frac{2\tilde{L}_{\mathcal{S}_n}}{s} + \frac{\gamma}{2}\cdot\max(0, \|\boldsymbol{W}\|-R)^2$, with $R \geq \max(1, 2r)$, where $r := \int \|\boldsymbol{W}\| d\mu_s$, and $0 < \gamma \leq \frac{1}{768T}$.

The proof for the above theorem is given in Appendix A.1. For completeness we also note in Appendix A.2 a form of distributional convergence for the LMC considered here that holds in the TV (Total Variation) metric and therein we point out the trade-offs with respect to the guarantees obtained above.

### 4.1 Sketch of the Proof of the (Main) Theorem 4.1

The first key step in the proof is to note that for a Gibbs measure that satisfies the PI, convergence to it in the 2-Rényi divergence implies convergence in Wasserstein ($W_2$) distance Liu (2020). Define the population risk as $\mathcal{R}(\boldsymbol{W}) = \mathbb{E}_{S_n}[\tilde{L}_{\mathcal{S}_n}(\boldsymbol{W})]$, where $S_n$ is sampled from the training distribution. Hence, as we can conclude convergence in $W_2$ from Theorem 4.2, we can leverage the argument in Raginsky et al. (2017) to demonstrate risk minimization through the following three steps:

(i) Directly applying Lemmas 3 and 6 from Raginsky et al. (2017), it can be shown that Theorem 4.2 further implies convergence in expectation of the population risk, evaluated over the distribution of the iterates, to the expected population risk under the stationary measure of the LMC i.e the Gibbs distribution of the empirical loss. (ii) By adapting the stability argument for the Gibbs measure (i.e., Proposition 12 of Raginsky et al. (2017)), it can be shown that for weights sampled from the Gibbs distribution the gap between the population risk and the empirical risk is inverse in the sample size. (iii) Using Proposition 11 from Raginsky et al. (2017), it can be shown that sampling from the Gibbs distribution is an approximate empirical risk minimizer for the losses we consider. Combining these 3 steps it can be shown that under LMC, the iterates converge to the minimum population risk of the neural losses considered here. A detailed proof is discussed in Appendix B.3.

## 5 Experiments

All experiments in this paper were conducted on neural networks as defined in Definition 1, specifically on depth-2 networks with a fixed last layer. The training data was generated from the function $2\sin(\pi x)$, where $x \in \left[-\frac{1}{2}, \frac{1}{2}\right]$. The outer layer norm $\|\boldsymbol{a}\|_2$ was fixed at 2. For the chosen tanh activation function, the constants

$M_D$ and $L$, as defined in Definition 2, can be both be set to 1. Additionally, $B_x$, as defined in Definition 1, can be set to $\frac{1}{2}$ given the specific data domain.

Substituting these values into the expression for the critical regularization parameter from Lemma 3.2, we obtain $\lambda_c^{\text{MSE}} = 2$. During training, we performed a grid-search over learning rates $[1e-3, 5e-3, 1e-2, 5e-2, 1e-1, 5e-1]$ for each width and selected the rate yielding the best performance. We also observed that setting $s$ to 1e−4 yielded the best performance.

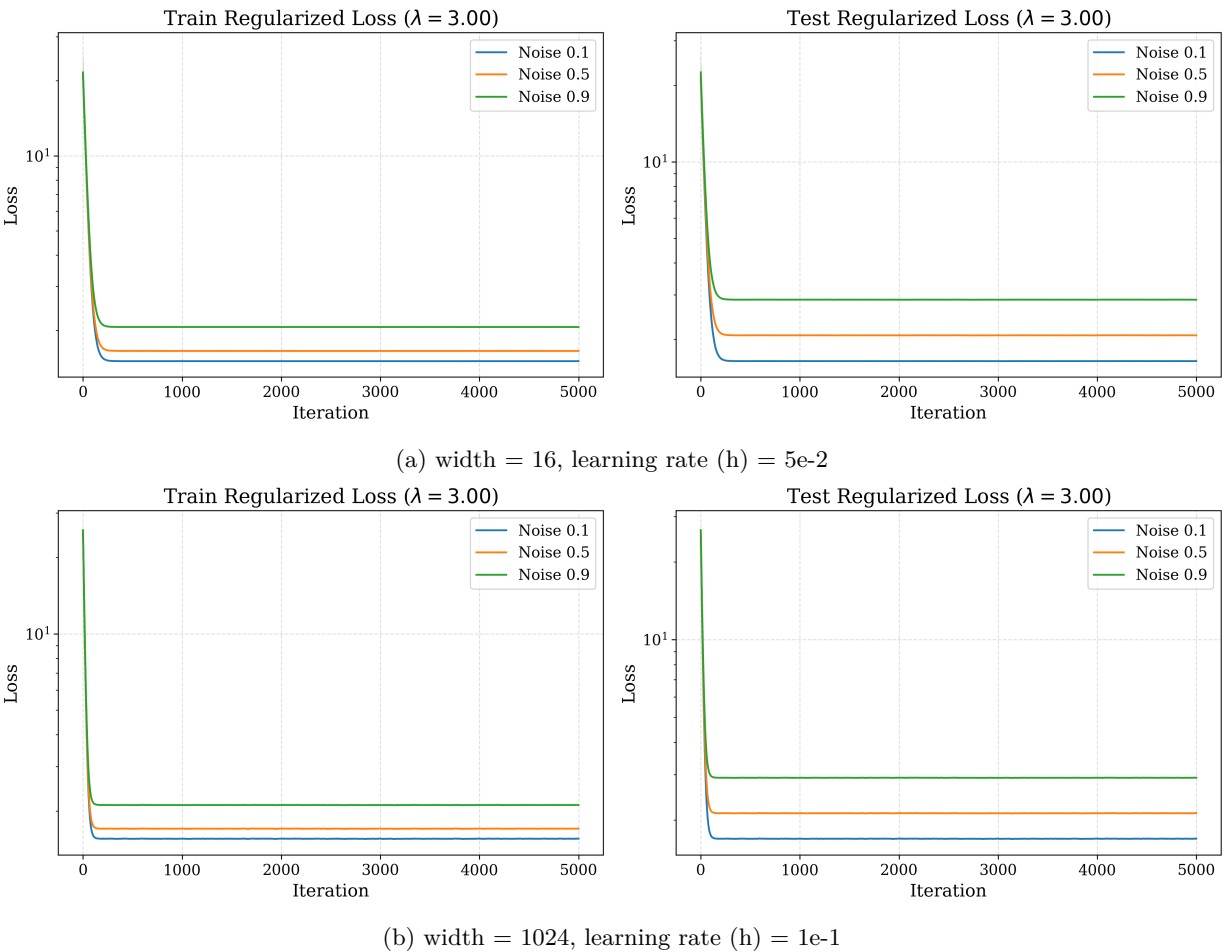

(a) width = 16, learning rate (h) = 5e-2

(b) width = 1024, learning rate (h) = 1e-1

Figure 2: This figure demonstrates that at widths 16 and 1024, the introduction of noise to the training and testing data substantially influences the final loss, indicating that the regularization parameter $\lambda$ does not dominate the regularized MSE loss being studied.

All experiments in here were conducted on CPUs, with each completing within 15 minutes. The codes can be found in this anonymized GitHub repository.

Figure 2 illustrates that at two different widths, with the regularization parameter $\lambda$ set just above the critical value discussed previously, the addition of noise significantly alters the minimum loss, indicating that the critical $\lambda$ is not excessively large. Here, the initial weights are sampled from a normal distribution of variance $\frac{1}{\text{width}}$.

Similar to Figure 1, Figure 3 also illustrates the phenomenon described in Theorem 4.1, with the only difference being that the initial weights are sampled from a normal distribution of variance $\frac{1}{\text{width}}$.

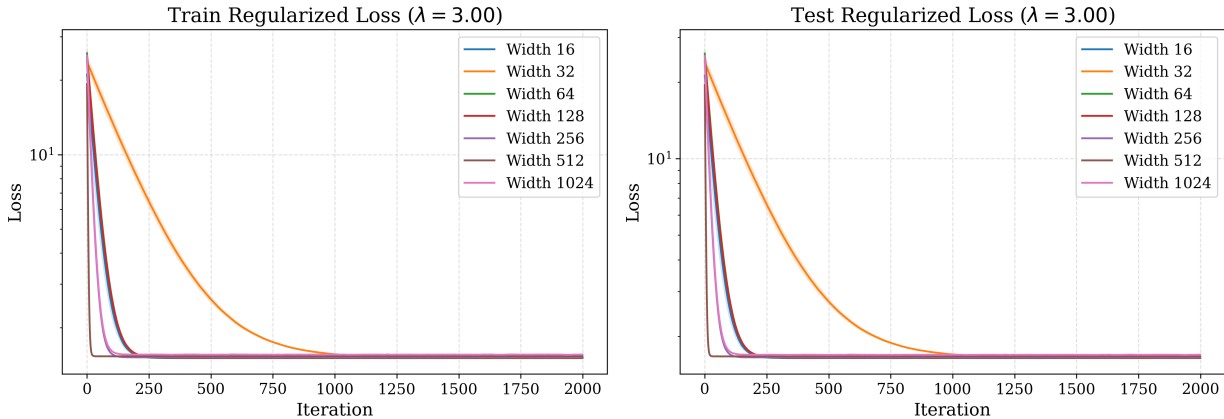

Figure 3: This figure presents the training and testing error plots for depth-2 neural networks with varying widths, where the initial weights are sampled from a normal distribution of variance $\frac{1}{\mathrm{width}}$.

## 5.1 Comparison with AdamW Optimizer

Figure 4 demonstrates that using the AdamW optimizer with a mini-match size of 16, and the same $\lambda$ as in the earlier LMC experiments, yields comparable performance.

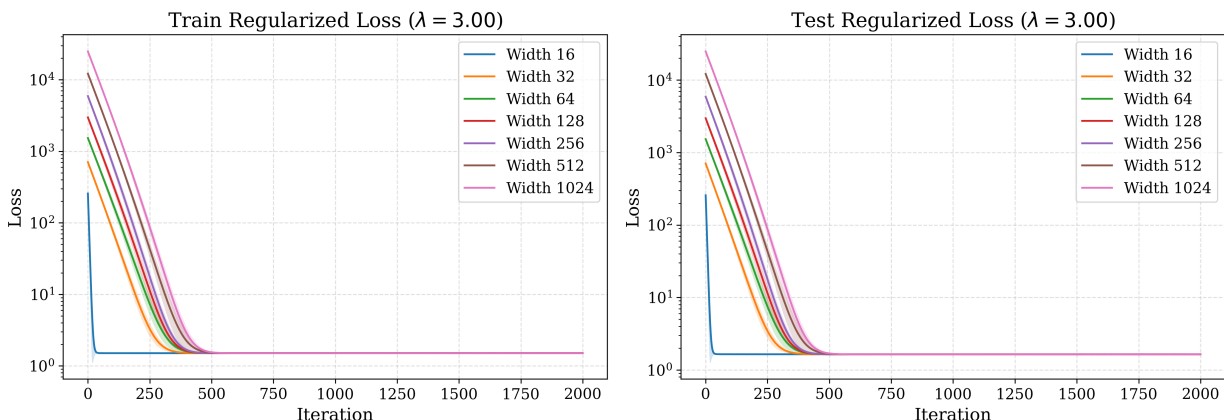

Figure 4: This figure presents the training and testing error plots for training depth-2 neural networks with varying widths, using AdamW optimizer for regression on the same data as above.

### 5.1.1 Comparison with AdamW Optimizer, Below the Villani Threshold

Figures 5a and 5b illustrate performance in a setting that is similar to that in Figures 1 and 4, respectively, with the only difference being that $\lambda$ is chosen below the Villani threshold. We observe that the performance of both optimizers continues to be comparable, which strengthens our argument that LMC can serve as an insightful theoretical model for optimizers deployed for neural training.

## 6 Discussion

We note that applying a perturbation argument known as Miclo's trick (Lemma 2.1 in Bardet et al. (2018)), one can argue that, since the loss functions we consider can be decomposed into two components — a strongly convex regularizer and a loss term that is Lipschitz continuous — the Gibbs measure of the loss function satisfies the LSI. However, the LSI constant $C_{LSI}$ is always larger than the Poincaré constant $C_{PI}$ that our current results involve Menz & Schlichting (2014). On the other hand, results of Chewi et al. (2024) reviewed earlier indicate that LSI potentials would have faster convergence times. We posit that a precise understanding of this trade-off can be an exciting direction of future research.

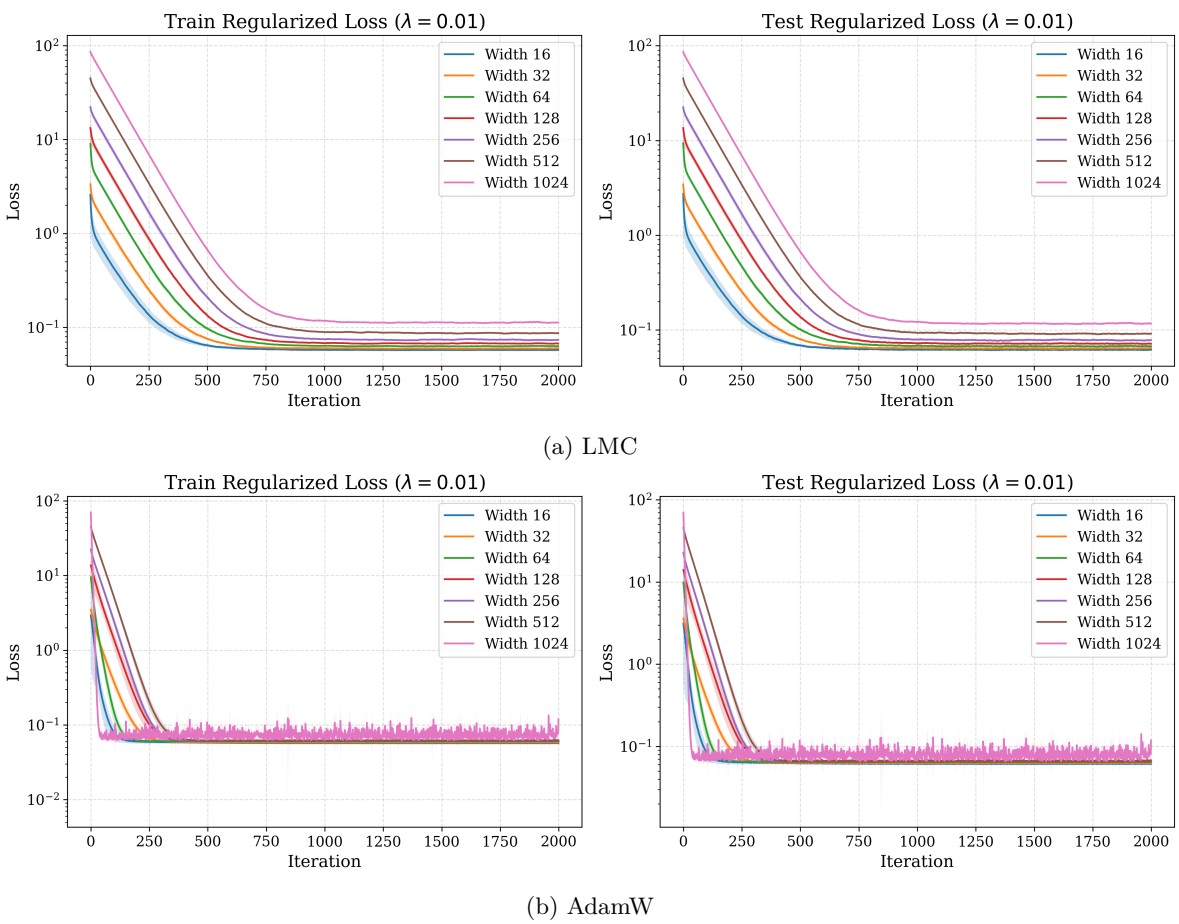

Figure 5: This figure presents the training and testing error plots for depth-2 neural networks with varying widths, using a value of $\lambda$ below the Villani threshold.

Going beyond gradient Lipschitz losses, Gopalani et al. (2024) and Gopalani & Mukherjee (2025), showed that two layer neural nets with SoftPlus activation function, defined as $\frac{1}{\beta}\ln(1 + \exp(\beta x))$ for some $\beta > 0$, can also satisfy the Villani conditions at similar thresholds of regularization as in the cases discussed here. As shown in Gopalani et al. (2024) and Gopalani & Mukherjee (2025), this leads to the conclusion that certain SDEs can converge exponentially fast to their empirical loss minima.

To put the above in context, we recall that for any diffusion process, the corresponding Gibbs measure must satisfy some isoperimetry inequality for convergence. However, the squared loss on SoftPlus activation is neither Hölder continuous, and because its not Lipschitz, nor can Miclo's trick be invoked on it to regularize it and induce isoperimetry for its Gibbs measure. *And yet, for squared loss on SoftPlus nets, one can show exponentially fast convergence of Langevin diffusion by arguing the needed isoperimetry via proving its regularized version to be a Villani function,* Gopalani & Mukherjee (2025). To the best of our knowledge there is no known alternative route to such a convergence. But it remains open to prove the convergence of any noisy gradient based discrete time algorithm for these nets.

Several other open questions get motivated from the possibilities uncovered in this work, some of which we list as follows. (a) It remains an open question whether PINN losses De Ryck & Mishra (2024) deployed for solving PDEs are Villani, in particular without explicit regularization — this could be possible because the PINN loss structure naturally allows for tunable regularization when enforcing boundary or initial conditions for the target PDE. (b) A very challenging question is to bound the Poincaré constants for the neural loss functions considered here, and thus gain more mathematical control on the run-time of LMC derived here.

We recall that understanding the distributional law of the asymptotic iterates is also motivated by the long standing need for uncertainty quantification of neural net training. So it gives further impetus to prove such results as given here for more general classes of neural losses than considered here.

For Gibbs measure of potentials with sub-linear or logarithmic tails that satisfy a weaker version of the Poincaré inequality, Mousavi-Hosseini et al. (2023) proved the convergence of LMC. This weak-PI condition can be asserted for much broader classes of neural networks. However, as noted in Mousavi-Hosseini et al. (2023), the weak-PI constant grows exponentially in dimension for Gibbs measure of potentials with logarithmic tails. We recall that generic upperbounds on the Poincaré constant are also exponential in dimension. Hence an interesting open question is whether there exists neural networks with a sub-exponential weak PI constant. Though, we note that a weak-PI based convergence via the results in Mousavi-Hosseini et al. (2023) are hard to interpret as they do not lead to a determination of the convergence time of the LMC as an explicit function of the target accuracy — as is the nature of the guarantees here via establishing of Villani conditions.

Lastly, we note that for convex potentials, Altschuler & Talwar (2022) established the first concentration bounds for LMC iterates. Such results are critical and remain open for any kind of neural net losses.

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

## A  Proofs of Main Theorems

### A.1  Proof of Theorem 4.2

*Proof of Theorem 4.2.* For the neural nets and data considered in Definition 1, by referring to Lemma 3.1 and 3.2 for the logistic loss and the squared loss scenarios, respectively, we can conclude that when the regularization parameter is set above the critical values $\lambda_c^{\text{BCE}}$ and $\lambda_c^{\text{MSE}}$ — as stated in the theorem statement — the corresponding losses are Villani functions (Definition 4).

From Theorem 3.3, it follows that the Gibbs measure of $\tilde{L}_{\mathcal{S}_n}$,

$$\mu_s = \frac{1}{Z_s} \exp\left\{-\frac{2\tilde{L}_{\mathcal{S}_n}}{s}\right\} \tag{5}$$

where $s > 0$, satisfies a Poincaré-type inequality for some $C_{PI} > 0$.

Now, by invoking Lemma 3.1 and 3.2 on the corresponding loss functions we obtain an upper-bound on the gradient-Lipschitz constant $\beta$ of $\tilde{L}_{\mathcal{S}_n}$.

For the LMC

$$\boldsymbol{W}_{(k+1)h} = \boldsymbol{W}_{kh} - h\nabla V(\boldsymbol{W}_{kh}) + \sqrt{2}(\boldsymbol{B}_{(k+1)h} - \boldsymbol{B}_{kh}), \tag{6}$$

if we define the objective function as,

$$V := \frac{2\tilde{L}_{\mathcal{S}_n}}{s}, \tag{7}$$

this LMC matches the LMC in Definition 5, which then matches the LMC in Chewi et al. (2024). The stationary measure from Theorem 7 of Chewi et al. (2024) then becomes our $\mu_s$, which satisfies the two conditions that are necessary for this theorem to hold i.e. it's gradient-Lipschitz with some constant $\beta$ and it satisfies a Poincaré-type inequality, as argued above.

Further, let's recall the continuous-time interpolation (Definition 6) and define the law of this continuous-time interpolation of LMC to be $\pi_t$ for some time step $t \geq 0$.

Recalling the definition of $r$ from the theorem statement, let's define a slightly modified measure $\hat{\mu}_s$ as

$$\hat{\mu}_s \propto \exp(-\hat{V}) ; \quad \hat{V} := \frac{2\tilde{L}_{\mathcal{S}_n}}{s} + \frac{\gamma}{2} \cdot \max(0, \|x\| - R)^2$$

where $R \geq \max(1, 2r)$ and $0 < \gamma \leq \frac{1}{768T}$, with $T = \tilde{\Theta}\left(qC_{PI}R_{2q-1}(\pi_0\|\mu_s)^{2/\alpha-1}\right)$. Then, from Theorem 7 of Chewi et al. (2024) ,we can say that the LMC with the following step-size,

$$h_q = \tilde{\Theta}\left(\frac{\varepsilon}{pdq^2 C_{PI}\ \tilde{\beta}(L_0, \beta)^2\ R_{2q-1}(\pi_0\|\mu_s)} \times \min\left\{1, \frac{1}{q\varepsilon}, \frac{pd}{r}, \frac{pd}{R_2(\pi_0\|\hat{\mu}_s)^{1/2}}\right\}\right),$$

satisfies $R_q(\pi_{N_q h_q}\|\mu_s) \leq \varepsilon$ for $q \geq 2$ after

$$N_q = \frac{T}{h_q} = \tilde{\Theta}\left(\frac{pdq^3 C_{PI}^2\ \tilde{\beta}(L_0, \beta)^2\ R_{2q-1}(\pi_0\|\mu_s)^2}{\varepsilon} \times \max\left\{1, q\varepsilon, \frac{r}{pd}, \frac{R_2(\pi_0\|\hat{\mu}_s)^{1/2}}{pd}\right\}\right)$$

$\square$

**Remark.** We recall from Lemma 31 and 32 of Chewi et al. (2024) that we can choose the initialization such that we can reasonably say that $R_2(\pi_0\|\hat{\mu}_s), R_{2q-1}(\pi_0\|\mu_s) = \tilde{O}(pd)$. Thus, Theorem 4.2 implies a convergence rate of $\tilde{O}\left(\frac{q^3 p^3 d^3 C_{PI}^2 \tilde{\beta}(L_0,\beta)^2}{\varepsilon} \times \max(1, \frac{r}{pd})\right)$.

## A.2 Convergence in TV of LMC on Depth 2 Neural Nets

Firstly, we note that in Corollary 8 of Balasubramanian et al. (2022), it was shown that for any measure $\mu \propto \exp(-V)$, where $V$ is gradient-Lipschitz and $\mu$ satisfies a Poincaré-type inequality, for a certain step-size, the average measure of the law of continuous-time interpolation of LMC converges to $\mu$ in TV. In the following, we show that the natural neural network setups considered in this work allow for invoking this result to get a similar distributional convergence of a stochastic neural training algorithm.

**Theorem A.1 (Convergence of LMC in TV for Appropriately Regularized Neural Nets).** Let $(\pi_t)_{t \geq 0}$ denote the law of continuous-time interpolation of LMC (Definition 6) with step-size $h > 0$, invoked on the objective function $\frac{2\tilde{L}_{\mathcal{S}_n}}{s}$, where $s > 0$ is an arbitrary scale constant and loss $\tilde{L}_{\mathcal{S}_n}$ being the regularized logistic or the squared loss on a depth-2 net as defined in Definition 1 with its regularization parameter being set above the critical value $\lambda_c^{\mathrm{BCE}}$ and $\lambda_c^{\mathrm{MSE}}$ as specified in Lemma 3.1 for the logistic loss and Lemma 3.2 for the squared loss.

We denote the Gibbs measure of the LMC objective function as $\mu_s \propto \exp\left\{\frac{-2\tilde{L}_{\mathcal{S}_n}}{s}\right\}$. If $D_{\mathrm{KL}}\left(\pi_0 \,\|\, \mu_s\right) \leq K_0$ and we choose the step-size $h = \frac{\sqrt{K_0}}{2\beta\sqrt{pdN}}$ then a certain averaged measure of the interpolated LMC $\overline{\pi}_{Nh} \coloneqq \frac{1}{Nh} \int_0^{Nh} \pi_t dt$ converges to the above Gibbs measure in total variation (TV) as follows,

$$\|\overline{\pi}_{Nh} - \mu_s\|_{\mathrm{TV}}^2 \coloneqq \left\| \frac{1}{Nh} \int_0^{Nh} \pi_t dt - \mu_s \right\|_{\mathrm{TV}}^2 \leq \frac{2C_{\mathrm{PI}}\beta\sqrt{pdK_0}}{\sqrt{N}} \tag{8}$$

In above $C_{\mathrm{PI}}$ is the Poincaré constant corresponding to the Gibbs measure $\mu_s$ satisfying a Poincaré-type inequality and $\beta$ is the gradient-Lipschitz constant of the loss function $\tilde{L}_{\mathcal{S}_n}$ i.e. $\beta_{\mathrm{MSE}}$ or $\beta_{\mathrm{BCE}}$ as given in Lemma 3.2 for the squared loss and Lemma 3.1 for the logistic loss, as the case maybe.

We note that, the convergence rate obtained by Theorem 4.2 has better dependence on $\varepsilon$ than Theorem A.1 but worse in the dimension $d$. Furthermore, the convergence in Theorem 4.2 is of the distribution of the last iterate while Theorem A.1 is in the average measure of the distribution of the iterates.

*Proof of Theorem A.1.* For the neural nets and data considered in Definition 1, by referring to Lemma 3.1 and 3.2 for the logistic loss and the squared loss scenarios, respectively, we can conclude that when the regularization parameter is set above the critical values $\lambda_c^{\mathrm{BCE}}$ and $\lambda_c^{\mathrm{MSE}}$ — as stated in the theorem statement — the corresponding losses are Villani functions (Definition 4).

From Theorem 3.3, it follows that that the Gibbs measure of $\tilde{L}_{\mathcal{S}_n}$,

$$\mu_s = \frac{1}{Z_s} \exp\left\{-\frac{2\tilde{L}_{\mathcal{S}_n}}{s}\right\} \tag{9}$$

where $s > 0$, satisfies a Poincaré-type inequality for some $C_{PI} > 0$.

Now, by invoking Lemma 3.1 and 3.2 on the corresponding loss functions we obtain an upper-bound on the gradient-Lipschitz constant $\beta$ of $\tilde{L}_{\mathcal{S}_n}$.

For the LMC

$$\boldsymbol{W}_{(k+1)h} = \boldsymbol{W}_{kh} - h\nabla V(\boldsymbol{W}_{kh}) + \sqrt{2}(\boldsymbol{B}_{(k+1)h} - \boldsymbol{B}_{kh}), \tag{10}$$

if we define the objective function as,

$$V \coloneqq \frac{2\tilde{L}_{\mathcal{S}_n}}{s}, \tag{11}$$

this LMC matches the LMC in Definition 5, which then matches the LMC in Balasubramanian et al. (2022). The measure from Corollary 8 of Balasubramanian et al. (2022) then becomes our $\mu_s$, which satisfies the two conditions that are necessary for the corollary to hold i.e. it's gradient-Lipschitz with some constant $\beta$ and it satisfies a Poincaré-type inequality, as argued above.

Further, let's recall the continuous-time interpolation (Definition 6) and define the law of this continuous-time interpolation of LMC to be $\pi_t$ for some time step $t \geq 0$ and its averaged measure $\overline{\pi}_{Nh}$, where,

$$\overline{\pi}_{Nh} \coloneqq \frac{1}{Nh} \int_0^{Nh} \pi_t dt \tag{12}$$

Then, from Corollary 8 of Balasubramanian et al. (2022), we obtain an upper-bound on the TV distance between $\overline{\pi}_{Nh}$ and $\mu_s$

$$\|\overline{\pi}_{Nh} - \mu_s\|_{\mathrm{TV}}^2 \leq \frac{2C_{\mathrm{PI}}\beta\sqrt{pdK_0}}{\sqrt{N}} \tag{13}$$

□

**Remark.** Theorem 4.2 achieves a better dependence on $\varepsilon$ but worse dependence on the dimension $d$ compared to Theorem A.1. Theorem 4.2 analyzes the last-iterate distribution, whereas Theorem A.1 focuses on the average measure of the iterates' distribution.

## B   Risk Minimization for Villani Nets under LMC

We begin with redefining the empirical loss as was given in Definition 1 in notation more appropriate for giving the proof of Theorem 4.1.

**Definition 7** (Loss Function). Recall the the depth-2 neural nets considered in this work, $\mathbb{R}^d \ni \boldsymbol{x} \mapsto f(\boldsymbol{x}; \boldsymbol{a}, \boldsymbol{W}) = \boldsymbol{a}^T \sigma(\boldsymbol{W}\boldsymbol{x}) \in \mathbb{R}$, where $\boldsymbol{a} \in \mathbb{R}^p$ and $\boldsymbol{W} = [\boldsymbol{w}_1, \ldots, \boldsymbol{w}_p]^\top \in \mathbb{R}^{p \times d}$. Correspondingly, we define the empirical loss function as $\tilde{L}_{\mathcal{S}_n}(\boldsymbol{W}) = \frac{1}{n} \sum_{i=1}^n \tilde{L}_i(\boldsymbol{W})$, where $\mathcal{S}_n \in X^n$ is the set of $n$ training data points, and $X$ is a random variable of some unknown distribution over $\mathbb{R}^d \times \mathbb{R}$. We further define the population risk as $\mathcal{R}(\boldsymbol{W}) \coloneqq \mathbb{E}_{\mathcal{S}_n}[\tilde{L}_{\mathcal{S}_n}(\boldsymbol{W})]$.

We will be considering two options for $\tilde{L}_i$,

1. (MSE loss) $\tilde{L}_i(\boldsymbol{W}) = \frac{1}{2}(y_i - f(\boldsymbol{x}_i; \boldsymbol{a}, \boldsymbol{W}))^2 + \frac{\lambda}{2}\|\boldsymbol{W}\|_F^2$

2. (BCE loss) $\tilde{L}_i(\boldsymbol{W}) = \log(1 + \exp\{-y_i f(\boldsymbol{x}_i; \boldsymbol{a}, \boldsymbol{W})\}) + \frac{\lambda}{2}\|\boldsymbol{W}\|_F^2$.

For any given $s > 0$, corresponding to the loss functions given above we define the Gibbs' measure as $\frac{1}{Z_s} \exp\left\{-\frac{2\tilde{L}_{\mathcal{S}_n}(\boldsymbol{W})}{s}\right\}$, where $Z_s$ is a normalization factor.

For Theorem 4.1, we define the following constant, which depends on the initial distribution

$$\kappa_0 \coloneqq \log \int_{\mathbb{R}^{p \times d}} e^{\|\boldsymbol{W}\|^2} p_0(\boldsymbol{W}) d\boldsymbol{W} < \infty,$$

and refer to the existence of such a constant as a "nice" initial distribution.

### B.1   Boundedness, Smoothness and Dissipativity of the Neural Losses

**Claim 1.** The function $\tilde{L}_i$ takes nonnegative real values and $\exists\ A, B \geq 0$ s.t. $\left|\tilde{L}_i(0)\right| \leq A$ and $\left\|\nabla \tilde{L}_i(0)\right\| \leq B\ \forall\ i \in [n]$.

*Proof.* For MSE loss, setting $\boldsymbol{W} = 0$ we get

$$\left|\tilde{L}_i(0)\right| \leq \left|\frac{1}{2}(y_i + |\langle \boldsymbol{a}, \boldsymbol{c}\rangle|)^2\right| \leq \left|\frac{(B_y + |\langle \boldsymbol{a}, \boldsymbol{c}\rangle|)^2}{2}\right| = A_{MSE}$$

Now, taking the gradient of the loss

$$\left\|\nabla_{\boldsymbol{w}_k} \tilde{L}_i(0)\right\| = \|(y_i - f(\boldsymbol{x}_i; \boldsymbol{a}, 0))\nabla_{\boldsymbol{w}_k} f(\boldsymbol{x}_i; \boldsymbol{a}, 0)\| \leq \|\boldsymbol{a}\|_2 B_x M_D(B_y + |\langle a, c\rangle|)$$

Concatenating for all $k$ we get

$$\left\|\nabla \tilde{L}_i(0)\right\| \leq \sqrt{p}\|\boldsymbol{a}\|_2 B_x M_D(B_y + |\langle a, c\rangle|) = B_{MSE}$$

Now, for BCE loss, setting $\boldsymbol{W} = 0$ we get

$$\left|\tilde{L}_i(0)\right| = \left|\log(1 + \exp\{-y_i \langle \boldsymbol{a}, \boldsymbol{c}\rangle\})\right| \le \left|\log(1 + \exp\{\langle \boldsymbol{a}, \boldsymbol{c}\rangle\})\right| = A_{BCE}$$

Now, taking the gradient of the loss

$$\left\|\nabla_{\boldsymbol{w}_k} \tilde{L}_i(0)\right\| = \left\|\nabla_{\boldsymbol{w}_k} \log(1 + \exp\{-y_i f(\boldsymbol{x}_i; \boldsymbol{a}, 0)\})\right\|_2 = \left\|\frac{-y_i}{1 + \exp\{-y_i f(\boldsymbol{x}_i; \boldsymbol{a}, 0)\}} \nabla_{\boldsymbol{w}_k} f(\boldsymbol{x}_i; \boldsymbol{a}, 0)\right\|_2$$

$$\le \|\boldsymbol{a}\|_2 B_x M_D \left\|\frac{1}{1 + \exp\{-y_i f(\boldsymbol{x}_i; \boldsymbol{a}, 0)\}}\right\|_2 \le \|\boldsymbol{a}\|_2 B_x M_D \left\|\frac{1}{1 + \exp\{-\langle \boldsymbol{a}, \boldsymbol{c}\rangle\}}\right\|_2$$

Concatenating for all $k$ we get

$$\left\|\nabla \tilde{L}_i(0)\right\| \le \sqrt{p} \|\boldsymbol{a}\|_2 B_x M_D \left\|\frac{1}{1 + \exp\{-\langle \boldsymbol{a}, \boldsymbol{c}\rangle\}}\right\|_2 = B_{BCE}.$$

$\square$

**Claim 2.** For each $i \in [n]$, the function $\tilde{L}_i(\cdot)$ is $\beta$–smooth, for some $\beta > 0$,

$$\left\|\nabla \tilde{L}_i(\boldsymbol{W}) - \nabla \tilde{L}_i(\boldsymbol{V})\right\| \le \beta \|\boldsymbol{W} - \boldsymbol{V}\|, \quad \forall \ \boldsymbol{W}, \boldsymbol{V} \in \mathbb{R}^{p \times d}.$$

*Proof.* For MSE loss,

$$\nabla \tilde{L}_i(\boldsymbol{W}) = \nabla \frac{1}{2}(y_i - f(\boldsymbol{x}_i; \boldsymbol{a}, \boldsymbol{W}))^2 + \nabla \frac{\lambda}{2} \|\boldsymbol{W}\|_F^2$$

Now, for $k \in [p]$ we can write

$$\boldsymbol{g}_k(\boldsymbol{W}) \coloneqq \left\|\nabla_{\boldsymbol{w}_k} \tilde{L}_i(\boldsymbol{W}) - \nabla_{\boldsymbol{v}_k} \tilde{L}_i(\boldsymbol{V})\right\|_2$$

$$= \left\|\nabla_{\boldsymbol{w}_k} \frac{1}{2}(y_i - f(\boldsymbol{x}_i; \boldsymbol{a}, \boldsymbol{W}))^2 - \nabla_{\boldsymbol{v}_k} \frac{1}{2}(y_i - f(\boldsymbol{x}_i; \boldsymbol{a}, \boldsymbol{V}))^2 + \nabla_{\boldsymbol{w}_k} \frac{\lambda}{2} \|\boldsymbol{W}\|_F^2 - \nabla_{\boldsymbol{v}_k} \frac{\lambda}{2} \|\boldsymbol{V}\|_F^2\right\|_2$$

$$= \left\|(y_i - f(\boldsymbol{x}_i; \boldsymbol{a}, \boldsymbol{W}))\nabla_{\boldsymbol{w}_k} f(\boldsymbol{x}_i; \boldsymbol{a}, \boldsymbol{W}) - (y_i - f(\boldsymbol{x}_i; \boldsymbol{a}, \boldsymbol{V}))\nabla_{\boldsymbol{v}_k} f(\boldsymbol{x}_i; \boldsymbol{a}, \boldsymbol{V}) + \lambda(\boldsymbol{w}_k - \boldsymbol{v}_k)\right\|_2$$

$$\le \left\|(y_i - f(\boldsymbol{x}_i; \boldsymbol{a}, \boldsymbol{W}))\nabla_{\boldsymbol{w}_k} f(\boldsymbol{x}_i; \boldsymbol{a}, \boldsymbol{W}) - (y_i - f(\boldsymbol{x}_i; \boldsymbol{a}, \boldsymbol{V}))\nabla_{\boldsymbol{v}_k} f(\boldsymbol{x}_i; \boldsymbol{a}, \boldsymbol{V})\right\|_2 + \lambda \|\boldsymbol{w}_k - \boldsymbol{v}_k\|_F$$

$$\le \|\boldsymbol{a}\|_2 B_x \|(y_i - f(\boldsymbol{x}_i; \boldsymbol{a}, \boldsymbol{W}))\sigma'(\langle \boldsymbol{w}_k, \boldsymbol{x}_i\rangle) - (y_i - f(\boldsymbol{x}_i; \boldsymbol{a}, \boldsymbol{V}))\sigma'(\langle \boldsymbol{v}_k, \boldsymbol{x}_i\rangle)\|_2 + \lambda \|\boldsymbol{w}_k - \boldsymbol{v}_k\|_F$$

So, this problem reduces to determining the Lipschitz constant of $F(\boldsymbol{W}) \coloneqq (y_i - f(\boldsymbol{x}_i; \boldsymbol{a}, \boldsymbol{W}))\sigma'(\langle \boldsymbol{w}_k, \boldsymbol{x}_i\rangle)$. This can be split as

$$F(\boldsymbol{W}) = \underbrace{y_i \sigma'(\langle \boldsymbol{w}_k, \boldsymbol{x}_i\rangle)}_{F_1} \underbrace{-f(\boldsymbol{x}_i; \boldsymbol{a}, \boldsymbol{W})\sigma'(\langle \boldsymbol{w}_k, \boldsymbol{x}_i\rangle)}_{F_2}$$

Now, looking at $F_2$, we can show that this is Lipschitz if it's gradient is bounded, to that end we take its gradient,

$$\left\|\nabla_{\boldsymbol{w}_j} F_2(\boldsymbol{W})\right\|_2 = \left\|\nabla_{\boldsymbol{w}_j}(f(\boldsymbol{x}_i; \boldsymbol{a}, \boldsymbol{W})\sigma'(\langle \boldsymbol{w}_k, \boldsymbol{x}_i\rangle))\right\|_2$$

$$= \|(a_j \sigma'(\langle \boldsymbol{w}_j, \boldsymbol{x}_i\rangle)\sigma'(\langle \boldsymbol{w}_k, \boldsymbol{x}_i\rangle)\boldsymbol{x}_i) + (\langle \boldsymbol{a}, \sigma(\boldsymbol{W}\boldsymbol{x}_i)\rangle \sigma''(\boldsymbol{w}_j \boldsymbol{x}_i)\boldsymbol{x}_i)\|_2$$

$$\le \|\boldsymbol{a}\|_2 M_D^2 B_x + \|\boldsymbol{a}\|_2 \sqrt{p} B_\sigma M_D' B_x = L_{prod}$$

The $\sqrt{p}$ comes from applying Cauchy-Schwarz. Now, we can concatenate them for $k = 1, \ldots, p$ to get,

$$\|F_2(\boldsymbol{W}) - F_2(\boldsymbol{V})\|_2 \le \|[L_{prod}(\boldsymbol{W}_1 - \boldsymbol{V}_1), \ldots, L_{prod}(\boldsymbol{W}_p - \boldsymbol{V}_p)]\|_2$$

$$\le \sqrt{p} L_{prod} \|[\boldsymbol{W} - \boldsymbol{V}]\|_2$$

The Lipschitz constant for $F_2(\boldsymbol{W})$ is $\sqrt{p}\|\boldsymbol{a}\|_2 M_D^2 B_x + \|\boldsymbol{a}\|_2 p B_\sigma M_D' B_x$. The Lipschitz constant for $F_2(\boldsymbol{W}) = y_i \sigma'(\langle \boldsymbol{w}_k, x_i \rangle)$ is $B_y L_\sigma'$. Now, using the fact that the Lipschitz constant of a sum of two functions is the sum of the Lipschitz constants we can say that the Lipschitz constant for $\boldsymbol{g}_k(\boldsymbol{W})$ for each $k \in [p]$ is

$$L_{row} = \|\boldsymbol{a}\|_2 B_x B_y L_\sigma' + \sqrt{p}\|\boldsymbol{a}\|_2^2 M_D^2 B_x^2 + p\|\boldsymbol{a}\|_2^2 B_x^2 M_D' B_\sigma + \lambda$$

Now, concatenating $\boldsymbol{g}_j(\boldsymbol{W})$ for each $j \in [p]$ we get

$$\nabla_{\boldsymbol{W}} \tilde{L}_i(\boldsymbol{W}) = \boldsymbol{g}(\boldsymbol{W}) \coloneqq [\boldsymbol{g}_1(\boldsymbol{W}), \ldots, \boldsymbol{g}_p(\boldsymbol{W})],$$

then the Lipschitz constant for $\boldsymbol{g}(\boldsymbol{W})$ is bounded as,

$$\beta = \mathrm{gLip}(\tilde{L}_i(\boldsymbol{W})) \le \sqrt{p}\left(\|\boldsymbol{a}\|_2 B_x B_y L_\sigma' + \sqrt{p}\|\boldsymbol{a}\|_2^2 M_D^2 B_x^2 + p\|\boldsymbol{a}\|_2^2 B_x^2 M_D' B_\sigma + \lambda\right).$$

Note that this upperbound matches the one in Lemma 3.2. A similar analysis of the upperbound on the gradient Lipschitz constant for the binary cross-entropy loss yields the same constant as in Lemma 3.1. $\quad\square$

**Claim 3.** For each $i \in [n]$, the function $\tilde{L}_i(\cdot)$ is $m, b$–dissipative, for some $m > 0$ and $b \ge 0$,

$$\left\langle \boldsymbol{W}, \nabla \tilde{L}_i(\boldsymbol{W}) \right\rangle \ge m\|\boldsymbol{W}\|^2 - b, \quad \forall \, \boldsymbol{W} \in \mathbb{R}^{p \times d}.$$

*Proof.* From Definition 7, for MSE loss we know,

$$\tilde{L}_i(\boldsymbol{W}) = \underbrace{\frac{1}{2}(y_i - f(\boldsymbol{x}_i; \boldsymbol{a}, \boldsymbol{W}))^2}_{L_{1,i}(\boldsymbol{W})} + \frac{\lambda}{2}\|\boldsymbol{W}\|_F^2$$

Taking the norm of the gradient of $L_{1,i}(\boldsymbol{W})$,

$$\begin{aligned}
\boldsymbol{g}_k(\boldsymbol{W}) \coloneqq \|\nabla_{\boldsymbol{w}_k} L_{1,i}(\boldsymbol{W})\| &= \left\|\nabla_{\boldsymbol{w}_k} \frac{1}{2}(y_i - f(\boldsymbol{x}_i; \boldsymbol{a}, \boldsymbol{W}))^2\right\|_2 \\
&= \|(y_i - f(\boldsymbol{x}_i; \boldsymbol{a}, \boldsymbol{W}))\nabla_{\boldsymbol{w}_k} f(\boldsymbol{x}_i; \boldsymbol{a}, \boldsymbol{W})\|_2 \le \|\boldsymbol{a}\|_2 B_x\|(y_i - f(\boldsymbol{x}_i; \boldsymbol{a}, \boldsymbol{W}))\sigma'(\langle \boldsymbol{w}_k, \boldsymbol{x}_i \rangle)\|_2 \\
&\le \|\boldsymbol{a}\|_2 B_x\|y_i \sigma'(\langle \boldsymbol{w}_k, \boldsymbol{x}_i \rangle)\|_2 + \|\boldsymbol{a}\|_2 B_x\|f(\boldsymbol{x}_i; \boldsymbol{a}, \boldsymbol{W})\sigma'(\langle \boldsymbol{w}_k, \boldsymbol{x}_i \rangle)\|_2 \\
&\le \|\boldsymbol{a}\|_2 B_x B_y M_D + B_x \sqrt{p}\|\boldsymbol{a}\|_2^2 B_\sigma M_D
\end{aligned}$$

Now, concatenating $\boldsymbol{g}_k(\boldsymbol{W})$ for each $k \in [p]$ we get,

$$\boldsymbol{g}(\boldsymbol{W}) \coloneqq [\boldsymbol{g}_1(\boldsymbol{W}), \ldots, \boldsymbol{g}_p(\boldsymbol{W})]$$

Then, the Lipschitz constant for $L_{1,i}(\boldsymbol{W})$ is bounded as,

$$\mathrm{Lip}(L_{1,i}(\boldsymbol{W})) \le \sqrt{p}\left(\|\boldsymbol{a}\|_2 B_x B_y M_D + B_x \sqrt{p}\|\boldsymbol{a}\|_2^2 B_\sigma M_D\right) = \alpha_{MSE}. \tag{14}$$

Now, for BCE loss we have,

$$\tilde{L}_i(\boldsymbol{W}) = \underbrace{\log(1 + \exp\{-y_i f(\boldsymbol{x}_i; \boldsymbol{a}, \boldsymbol{W})\})}_{L_{1,i}(\boldsymbol{W})} + \frac{\lambda}{2}\|\boldsymbol{W}\|_F^2$$

And for the corresponding gradient we have,

$$\begin{aligned}
\boldsymbol{g}_k(\boldsymbol{W}) \coloneqq \|\nabla_{\boldsymbol{w}_k} L_{1,i}(\boldsymbol{W})\| &= \|\nabla_{\boldsymbol{w}_k} \log(1 + \exp\{-y_i f(\boldsymbol{x}_i; \boldsymbol{a}, \boldsymbol{W})\})\|_2 \\
&= \left\|\frac{-y_i \exp\{-y_i f(\boldsymbol{x}_i; \boldsymbol{a}, \boldsymbol{W})\}}{1 + \exp\{-y_i f(\boldsymbol{x}_i; \boldsymbol{a}, \boldsymbol{W})\}} \nabla_{\boldsymbol{w}_k} f(\boldsymbol{x}_i; \boldsymbol{a}, \boldsymbol{W})\right\|_2 = \left\|\frac{-y_i}{1 + \exp\{y_i f(\boldsymbol{x}_i; \boldsymbol{a}, \boldsymbol{W})\}} \nabla_{\boldsymbol{w}_k} f(\boldsymbol{x}_i; \boldsymbol{a}, \boldsymbol{W})\right\|_2 \\
&\le \|\boldsymbol{a}\|_2 B_x \left\|\frac{1}{1 + \exp\{y_i f(\boldsymbol{x}_i; \boldsymbol{a}, \boldsymbol{W})\}} \sigma'(\langle \boldsymbol{w}_k, \boldsymbol{x}_i \rangle)\right\|_2 \le \|\boldsymbol{a}\|_2 B_x M_D \left\|\frac{1}{1 + \exp\{y_i f(\boldsymbol{x}_i; \boldsymbol{a}, \boldsymbol{W})\}}\right\|_2 \tag{15}
\end{aligned}$$

To upperbound the above term we need to perform the following simplification,

$$\frac{1}{1 + \exp\{y_i f(\boldsymbol{x}_i; \boldsymbol{a}, \boldsymbol{W})\}} \leq \frac{1}{1 + \exp\{-|f(\boldsymbol{x}_i; \boldsymbol{a}, \boldsymbol{W})|\}} = \frac{\exp\{|f(\boldsymbol{x}_i; \boldsymbol{a}, \boldsymbol{W})|\}}{1 + \exp\{|f(\boldsymbol{x}_i; \boldsymbol{a}, \boldsymbol{W})|\}}$$

Let's consider the function $h(z) = \frac{e^z}{1 + e^z}$, $z \in [0, \infty)$

$$h(0) = \frac{1}{2}, \quad h'(z) = \frac{e^z}{(1 + e^z)^2} \leq \frac{1}{4}$$

Hence, we have,

$$h(z) - h(0) = \int_0^z h'(z) \leq \int_0^z \frac{1}{4} = \frac{z}{4}$$

Therefore,

$$h(z) \leq \frac{1}{2} + \frac{z}{4}$$

Therefore, $\boldsymbol{g}_k(\boldsymbol{W})$ can be upper bounded as,

$$\boldsymbol{g}_k(\boldsymbol{W}) \leq \|\boldsymbol{a}\|_2 B_x M_D \left\| \frac{1}{2} + \frac{|f(\boldsymbol{x}_i; \boldsymbol{a}, \boldsymbol{W})|}{4} \right\|_2$$

$$\leq \|\boldsymbol{a}\|_2 B_x M_D \left( \frac{\sqrt{p}}{2} + \frac{\sqrt{p}\|\boldsymbol{a}\|_2 B_\sigma B_x}{4} \right)$$

Now, concatenating $\boldsymbol{g}_k(\boldsymbol{W})$ for each $k \in [p]$, the Lipschitz constant for $L_{1,i}(\boldsymbol{W})$ of the BCE loss is bounded as,

$$\mathrm{Lip}(L_{1,i}(\boldsymbol{W})) \leq \sqrt{p}\|\boldsymbol{a}\|_2 B_x M_D \left( \frac{\sqrt{p}}{2} + \frac{\sqrt{p}\|\boldsymbol{a}\|_2 B_\sigma B_x}{4} \right) = \alpha_{BCE}$$

As we have now shown that for both MSE and BCE loss $L_{1,i}(\boldsymbol{W})$ is $\alpha$-Lipschitz to prove that $\tilde{L}_i(\boldsymbol{W}) = L_{1,i}(\boldsymbol{W}) + \frac{\lambda}{2}\|\boldsymbol{W}\|_F^2$ is $m, b$-dissipative we can simplify it as follows,

$$\langle \boldsymbol{W}, \nabla \tilde{L}_i(\boldsymbol{W}) \rangle = \langle \boldsymbol{W}, \nabla_{\boldsymbol{W}} L_{1,i}(\boldsymbol{W}) + \lambda \boldsymbol{W} \rangle = \langle \boldsymbol{W}, \nabla_{\boldsymbol{W}} L_{1,i}(\boldsymbol{W}) \rangle + \langle \boldsymbol{W}, \lambda \boldsymbol{W} \rangle$$

$$= \langle \boldsymbol{W}, \nabla_{\boldsymbol{W}} L_{1,i}(\boldsymbol{W}) \rangle + \lambda \|\boldsymbol{W}\|^2$$

Now, by applying Cauchy-Schwarz and the fact that $\|\nabla_{\boldsymbol{W}} L_{1,i}(\boldsymbol{W})\| \leq \alpha$, we can say that

$$\langle \boldsymbol{W}, \nabla \tilde{L}_i(\boldsymbol{W}) \rangle \geq -\alpha\|\boldsymbol{W}\| + \lambda\|\boldsymbol{W}\|^2 = \frac{\lambda}{2}\left( 2\|\boldsymbol{W}\|^2 - 2\frac{\alpha}{\lambda}\|\boldsymbol{W}\| \right)$$

$$= \frac{\lambda}{2}\left( 2\|\boldsymbol{W}\|^2 - 2\frac{\alpha}{\lambda}\|\boldsymbol{W}\| + \frac{\alpha^2}{\lambda^2} - \frac{\alpha^2}{\lambda^2} \right) = \frac{\lambda}{2}\left( 2\|\boldsymbol{W}\|^2 - 2\frac{\alpha}{\lambda}\|\boldsymbol{W}\| + \frac{\alpha^2}{\lambda^2} \right) - \frac{\alpha^2}{2\lambda}$$

$$= \frac{\lambda}{2}\left( \sqrt{2}\|\boldsymbol{W}\| - \frac{\alpha}{\lambda} \right)^2 - \frac{\alpha^2}{2\lambda}$$

Thus, we can say that the two losses defined in Definition 7 are $m, b$-dissipative with $m = \frac{\lambda}{2}$ and $b = \frac{\alpha^2}{2\lambda}$. $\qquad \square$

## B.2 Intermediate Results Towards the Proof of Theorem 4.1

**Lemma B.1** (**Lemma 6 of Raginsky et al. (2017)**)**.** Let $\mu, \nu$ be two probability measures on $\mathbb{R}^d$ with finite second moments, and let $g : \mathbb{R}^{p \times d} \to \mathbb{R}$ be a $C^1$ function such that

$$\|\nabla g(\boldsymbol{W})\| \leq c_1\|\boldsymbol{W}\| + c_2, \quad \forall \boldsymbol{W} \in \mathbb{R}^{p \times d}$$

for some constants $c_1 > 0$ and $c_2 \geq 0$. Then

$$\left| \int_{\mathbb{R}^{p \times d}} g \, d\mu - \int_{\mathbb{R}^{p \times d}} g \, d\nu \right| \leq (c_1 \sigma + c_2) \mathcal{W}_2(\mu, \nu) \tag{16}$$

where $\sigma^2 := \max\left( \int_{\mathbb{R}^{p \times d}} \mu(d\boldsymbol{W})\|\boldsymbol{W}\|^2, \int_{\mathbb{R}^{p \times d}} \nu(d\boldsymbol{W})\|\boldsymbol{W}\|^2 \right)$.

We note that the above is a special case of Proposition 1 in Polyanskiy & Wu (2016).

The Stochastic Gradient Langevin Dynamics (SGLD) algorithm (Equation 1.3 of Raginsky et al. (2017)) reduces to the LMC algorithm of Definition 5 by setting $\eta = \frac{2h}{s}$ and $\beta = \frac{2}{s}$ and when the gradients are exact. The continuous-time diffusion process $d\boldsymbol{W}(t) = -\nabla \tilde{L}_{\mathcal{S}_n}(\boldsymbol{W}(t)) + \sqrt{s}, d\boldsymbol{B}(t)$ is obtained from Equation 1.4 of Raginsky et al. (2017) by setting $\beta = \frac{2}{s}$. Proposition 8 of Raginsky et al. (2017) shows that the law of the SGLD iterates is close to that of the corresponding continuous-time diffusion in 2-Wasserstein distance, and thus LMC can be regarded as a discretization of this diffusion process.

In the following lemma, we upper bound the second moments of the iterates of both the continuous-time and discrete-time processes.

**Lemma B.2** (**Lemma 3 of Raginsky et al. (2017)**). For all $0 < \frac{2h}{s} < \min\left(1, \frac{m}{4\beta^2}\right)$ and all $\boldsymbol{z} \in Z^n$

$$\sup_{k \geq 0} \mathbb{E}_z \|\boldsymbol{W}_{kh}\|^2 \leq \kappa_0 + 2 \cdot \max\left(1, \frac{1}{m}\right)\left(b + 2B^2 + \frac{pds}{2}\right)$$

and

$$\mathbb{E}_z \|\boldsymbol{W}(t)\|^2 \leq \kappa_0 e^{-2mt} + \frac{b + pds/2}{m}\left(1 - e^{-2mt}\right)$$
$$\leq \kappa_0 + \frac{b + pds/2}{m}$$

where $\boldsymbol{W}(t)$ are the iterates of the continuous time diffusion process $d\boldsymbol{W}(t) = -\nabla \tilde{L}_{\mathcal{S}_n}(\boldsymbol{W}(t)) + \sqrt{s}, d\boldsymbol{B}(t)$.

**Lemma B.3** (**Stability of Gibbs' Algorithm under PI**). Let $\mathcal{S}_n, \bar{\mathcal{S}}_n \in X^n$ be $n$ training data points sampled from $X^n$, where $\mathcal{S}_n$ and $\bar{\mathcal{S}}_n$ differ only at one index $i$. Now, let $\mu_{\mathcal{S}_n}$ and $\mu_{\bar{\mathcal{S}}_n}$ be the corresponding Gibbs' measure of the loss functions, i.e $\frac{1}{Z_s}\exp\left\{-\frac{2\tilde{L}_{\mathcal{S}_n}(\boldsymbol{W})}{s}\right\}$ and $\frac{1}{\bar{Z}_s}\exp\left\{-\frac{2\tilde{L}_{\bar{\mathcal{S}}_n}(\boldsymbol{W})}{s}\right\}$ respectively, where $s > 0$. If $\mu_{\bar{\mathcal{S}}_n}$ satisfy PI with some constant $C_{PI} > 0$, then

$$W_2(\mu_{\mathcal{S}_n}, \mu_{\bar{\mathcal{S}}_n}) \leq \frac{8 C_{PI}\sqrt{\mathcal{C}(\tilde{L}_{\mathcal{S}_n})}}{sn}\sqrt{B^2 + \frac{\beta^2(b + spd/2)}{m}}$$

where $\tilde{L}_{\mathcal{S}_n}$ is as given in Definition 7, $\tilde{L}_i(\boldsymbol{W})$ is gradient Lipschitz with constant $\beta$ by Claim 2, $\tilde{L}_i(\boldsymbol{W})$ is $(m, b)$-dissipative by Claim 3, $B$ bounds $\|\nabla \tilde{L}_i(\boldsymbol{W})\|$ by Claim 1, and $\mathcal{C}(\tilde{L}_{\mathcal{S}_n})$ is a constant that depends on the loss function.

*Proof of Lemma B.3.* From Theorem 1.1 of Liu (2020) we know that if a measure $\pi$ satisfies PI then $W_2^2(\mu, \pi) \leq 2C_{PI}\text{Var}_\pi(p)$ which in turn implies $W_2^2(\mu, \pi) \leq 2C_{PI}^2\mathbb{E}[\|\nabla p\|^2]$ where $p \coloneqq \frac{d\mu}{d\pi}$ is the Radon-Nikodym derivative.

Here, since $\mathcal{S}_n$ and $\bar{\mathcal{S}}_n$ are different only at $i$, we have

$$p(\boldsymbol{W}) = \frac{d\mu_{\mathcal{S}_n}}{d\mu_{\bar{\mathcal{S}}_n}} \coloneqq \frac{\exp\left(\frac{-2}{s}\left(\frac{1}{n}\sum_{i=1}^n \tilde{L}_i(\boldsymbol{W}) + \frac{\lambda}{2}\|\boldsymbol{W}\|_F^2 - \frac{1}{n}\sum_{i=1}^n \tilde{L}_i'(\boldsymbol{W}) - \frac{\lambda}{2}\|\boldsymbol{W}\|_F^2\right)\right)}{K}$$
$$= \frac{\exp\left(\frac{-2}{ns}(\tilde{L}_i(\boldsymbol{W}) - \tilde{L}_i'(\boldsymbol{W}))\right)}{K},$$

where $K = \frac{Z_s}{\bar{Z}_s}$ is a constant and $i \in [n]$ is the position where the two are different. Then

$$\nabla p(\boldsymbol{W}) = \frac{2}{ns}\left(\nabla_{\boldsymbol{W}}\tilde{L}_i'(\boldsymbol{W}) - \nabla_{\boldsymbol{W}}\tilde{L}_i(\boldsymbol{W})\right)p(\boldsymbol{W})$$

Now, using the above we can say that

$$W_2^2(\mu_{\mathcal{S}_n}, \mu_{\bar{\mathcal{S}}_n}) \le 2C_{PI}^2 \mathbb{E}[\|\nabla p(\boldsymbol{W})\|^2]$$

$$= \frac{8C_{PI}^2}{s^2 n^2} \int_{\mathbb{R}^{p \times d}} \left\|\nabla_{\boldsymbol{W}} \tilde{L}_i'(\boldsymbol{W}) - \nabla_{\boldsymbol{W}} \tilde{L}_i(\boldsymbol{W})\right\|^2 p^2(\boldsymbol{W}) d\mu_{\bar{\mathcal{S}}_n}$$

$$= \frac{8C_{PI}^2}{s^2 n^2} \int_{\mathbb{R}^{p \times d}} \left\|\nabla_{\boldsymbol{W}} \tilde{L}_i'(\boldsymbol{W}) - \nabla_{\boldsymbol{W}} \tilde{L}_i(\boldsymbol{W})\right\|^2 p(\boldsymbol{W}) d\mu_{\mathcal{S}_n}$$

Replacing $p(\boldsymbol{W})$ by a constant $\mathcal{C}(\tilde{L}_{\mathcal{S}_n})$ that upperbounds it, i.e. $\mathcal{C}(\tilde{L}_{\mathcal{S}_n}) \coloneqq \sup_{\boldsymbol{W} \in \mathbb{R}^{p \times d}} p(\boldsymbol{W})$,

$$\le \frac{8C_{PI}^2 \mathcal{C}(\tilde{L}_{\mathcal{S}_n})}{s^2 n^2} \int_{\mathbb{R}^{p \times d}} \left\|\nabla_{\boldsymbol{W}} \tilde{L}_i'(\boldsymbol{W}) - \nabla_{\boldsymbol{W}} \tilde{L}_i(\boldsymbol{W})\right\|^2 d\mu_{\mathcal{S}_n}$$

$$\le \frac{64C_{PI}^2 \mathcal{C}(\tilde{L}_{\mathcal{S}_n})}{s^2 n^2} \left(\beta^2 \int_{\mathbb{R}^{p \times d}} \|\boldsymbol{W}\|^2 d\mu_{\mathcal{S}_n} + B^2\right)$$

where $\mathcal{C}(\tilde{L}_{\mathcal{S}_n})$ is a constant that depends on the loss $\tilde{L}_{\mathcal{S}_n}(\cdot)$. Therefore,

$$W_2(\mu_{\mathcal{S}_n}, \mu_{\bar{\mathcal{S}}_n}) \le \frac{8C_{PI}\sqrt{\mathcal{C}(\tilde{L}_{\mathcal{S}_n})}}{sn} \sqrt{\beta^2 \int_{\mathbb{R}^{p \times d}} \|\boldsymbol{W}\|^2 d\mu_{\mathcal{S}_n} + B^2}$$

since the second moment of the weights are bounded in Lemma B.2 by $\frac{(b + spd/2)}{m}$ as $t \to \infty$, hence we can further simplify the above as

$$\le \frac{8C_{PI}\sqrt{\mathcal{C}(\tilde{L}_{\mathcal{S}_n})}}{sn} \sqrt{B^2 + \frac{\beta^2(b + spd/2)}{m}}$$

$\square$

**Lemma B.4 (Upper Bound on the Radon-Nikodym Derivative of Gibbs Measures Differing at One Point).** Let the loss function $\tilde{L}_{\mathcal{S}_n}$ be as in Definition 7. Then we can say that $\mathcal{C}(\tilde{L}_{\mathcal{S}_n}) \coloneqq \sup_{\boldsymbol{W} \in \mathbb{R}^{p \times d}} \frac{d\mu_{\mathcal{S}_n}}{d\mu_{\bar{\mathcal{S}}_n}}$, where $\mu_{\mathcal{S}_n}$ and $\mu_{\bar{\mathcal{S}}_n}$ are the Gibbs' measure for any two $\mathcal{S}_n, \bar{\mathcal{S}}_n \in X^n$ that differ only in a single coordinate, is upper-bounded by

1. $\frac{1}{K} \cdot \exp\left(\frac{2}{sn}\left(\frac{1}{2}\left(B_y + pa_{max}B_\sigma\right)^2\right)\right)$,
   for MSE loss, where $|y_i| \le B_y$, $a_{max} \coloneqq \max_{i \in [p]} |a_i|$ and $|\sigma(\cdot)| \le B_\sigma$, and

2. $\frac{1}{K} \cdot \exp\left(\frac{2}{sn}\left(\frac{1}{2}\left[\log\left(\frac{1 + \exp\{pa_{max}B_\sigma\}}{1 + \exp\{-pa_{max}B_\sigma\}}\right)\right]\right)\right)$,
   for BCE loss, where $a_{max} \coloneqq \max_{i \in [p]} |a_i|$ and $|\sigma(\cdot)| \le B_\sigma$,

where $K = \frac{Z_s}{\bar{Z}_s}$ as defined is Lemma B.3.

*Proof.* For MSE loss,

$$\left|\tilde{L}_i(\boldsymbol{W}) - \tilde{L}_i'(\boldsymbol{W})\right| = \frac{1}{2}\left[(y_i - f(\boldsymbol{x}_i; \boldsymbol{a}, \boldsymbol{W}))^2 - (\bar{y}_i - f(\bar{\boldsymbol{x}}_i; \boldsymbol{a}, \boldsymbol{W}))^2\right]$$

$$= \frac{1}{2}\left[(y_i - f(\boldsymbol{x}_i; \boldsymbol{a}, \boldsymbol{W}))^2\right]$$

$$\le \frac{1}{2}\left(B_y + pa_{max}B_\sigma\right)^2$$

Now, for BCE loss,

$$\left|\tilde{L}_i(\boldsymbol{W}) - \tilde{L}_i'(\boldsymbol{W})\right| \le \frac{1}{2}\left[\log\left(\frac{1 + \exp\{pa_{max}B_\sigma\}}{1 + \exp\{-pa_{max}B_\sigma\}}\right)\right].$$

Now from the definition of $\mathcal{C}(\tilde{L}_{\mathcal{S}_n}) = \sup_{\boldsymbol{W} \in \mathbb{R}^{p \times d}} \frac{d\mu_{\mathcal{S}_n}}{d\mu_{\bar{\mathcal{S}}_n}} = \sup_{\boldsymbol{W} \in \mathbb{R}^{p \times d}} \frac{\exp\left(\frac{-2}{ns}(\tilde{L}_i(\boldsymbol{W}) - \tilde{L}_i'(\boldsymbol{W}))\right)}{K}$, we can upperbound it by replacing $-(\tilde{L}_i(\boldsymbol{W}) - \tilde{L}_i'(\boldsymbol{W}))$ by its upperbound. $\square$

**Proposition B.5** (**Uniform Stability of Gibbs under PI**)**.** For any two $\mathcal{S}_n, \bar{\mathcal{S}}_n \in X^n$ that differ only in a single coordinate,

$$\sup_{\boldsymbol{x}_i \in X} \left| \int_{\mathbb{R}^{p \times d}} \tilde{L}_i(\boldsymbol{W}) \mu_{\mathcal{S}_n}(d\boldsymbol{W}) - \int_{\mathbb{R}^{p \times d}} \tilde{L}_i(\boldsymbol{W}) \mu_{\bar{\mathcal{S}}_n}(d\boldsymbol{W}) \right| \le \frac{\tilde{C}_3}{n}$$

where

$$\tilde{C}_3 \coloneqq 16\sqrt{2} \left( \beta^2 \frac{b + spd/2}{m} + B^2 \right) \frac{C_{PI} \sqrt{\mathcal{C}(\tilde{L}_{\mathcal{S}_n})}}{s}$$

and $\mathcal{C}(\tilde{L}_{\mathcal{S}_n})$ is upper-bounded as in Lemma B.4.

*Proof of Proposition B.5.* Since $\tilde{L}_{\mathcal{S}_n}$ satisfies the conditions of Lemma B.1 with $c_1 = \beta$ and $c_2 = B$, and $\mu_{\mathcal{S}_n}$ and $\mu_{\bar{\mathcal{S}}_n}$ satisfies Lemma B.1 with $\sigma^2 = \frac{b+spd/2}{m}$, which is a bound for the second moment of the probability measures that we obtain from Lemma B.2, we can say that

$$\sup_{\boldsymbol{x}_i \in X} \left| \int_{\mathbb{R}^{p \times d}} \tilde{L}_i(\boldsymbol{W}) \mu_{\mathcal{S}_n}(d\boldsymbol{W}) - \int_{\mathbb{R}^{p \times d}} \tilde{L}_i(\boldsymbol{W}) \mu_{\bar{\mathcal{S}}_n}(d\boldsymbol{W}) \right| \le 16\sqrt{2} \left( \beta^2 \frac{b + spd/2}{m} + B^2 \right) \frac{C_{PI} \sqrt{\mathcal{C}(\tilde{L}_{\mathcal{S}_n})}}{ns}$$

$\square$

**Lemma B.6** (2-**Rényi Upper Bounds** 2-**Wasserstein under PI**)**.** Let $\mu, \pi$ be two probability measures where $\mu$ satisfies the PI with constant $C_{PI}$, then

$$\mathcal{W}_2(\pi, \mu) \le 2C_{PI}(e^{R_2(\pi \| \mu)} - 1)$$

*Proof.* Since $\mu$ satisfies PI, we know from Theorem 1.1 of Liu (2020) that $\mathcal{W}_2(\pi, \mu) \le 2C_{PI} \text{Var}_\mu(f)$ where $f = \frac{d\pi}{d\mu}$. Then

$$\mathcal{W}_2(\pi, \mu) = 2C_{PI} \text{Var}_\mu(f) = 2C_{PI}(\mathbb{E}_\mu[f^2] - (\mathbb{E}_\mu[f])^2) = 2C_{PI}(\mathbb{E}_\mu[f^2] - 1) = 2C_{PI}(e^{R_2(\pi \| \mu)} - 1).$$

$\square$

For completeness we restate Proposition 11 from Raginsky et al. (2017) with revised notation,

**Proposition B.7** (**Almost-ERM property of the Gibbs' Algorithm (Proposition 11 Raginsky et al. (2017))**)**.** Given $\tilde{L}_{\mathcal{S}_n}(\boldsymbol{W})$ as in Definition 7, and it satisfies Claims 2 and 3, for any $s \le m$ we have

$$\int_{\mathbb{R}^{p \times d}} \tilde{L}_{Q_n}(\boldsymbol{W}) \mu_{\mathcal{S}_n}(d\boldsymbol{W}) - \min_{\boldsymbol{W} \in \mathbb{R}^{p \times d}} \tilde{L}_{\mathcal{S}_n}(\boldsymbol{W}) \le \frac{spd}{4} \log\left( \frac{e\beta}{m} \left( \frac{2b}{spd} + 1 \right) \right),$$

where $\tilde{L}_{Q_n}(\boldsymbol{W})$ is the loss evaluated over $Q_n = (\boldsymbol{x}_1, \dots, \boldsymbol{x}_n) \sim \mathbb{P}^{\otimes n}$ fixed set of $n$ data points sampled from the joint distribution $\mathbb{P}^{\otimes n}(d\mathcal{S}_n)$.

*Proof.*

$$\int_{\mathbb{R}^{p \times d}} \tilde{L}_{Q_n} \mu_{\mathcal{S}_n}(d\boldsymbol{W}) = \frac{s}{2} \left( \underbrace{- \int_{\mathbb{R}^{p \times d}} \frac{\exp\left\{ -\frac{2}{s} \tilde{L}_{Q_n}(\boldsymbol{W}) \right\}}{Z_s} \log \frac{\exp\left\{ -\frac{2}{s} \tilde{L}_{Q_n}(\boldsymbol{W}) \right\}}{Z_s} d\boldsymbol{W}}_{h_1} - \log Z_s \right) \tag{17}$$

From Theorem 4.2 and Lemma B.6 we know that $\mathcal{W}_2(\pi_{\mathcal{S}_n, N}, \mu_{\mathcal{S}_n}) \xrightarrow{N \to \infty} 0$. Since convergence in $\mathcal{W}_2$ is equivalent to weak convergence plus convergence in second moment (Villani (2003), Theorem 7.12), we have by Lemma B.2

$$\int_{\mathbb{R}^{p \times d}} \|\boldsymbol{W}\|^2 \mu_{\mathcal{S}_n}(d\boldsymbol{W}) = \lim_{N \to \infty} \int_{\mathbb{R}^{p \times d}} \|\boldsymbol{W}\|^2 \pi_{\mathcal{S}_n, N}(d\boldsymbol{W}) \le \frac{b + spd/2}{m}.$$

We can upperbound $h_1$, which is also known as the differential entropy of the probability measure $\frac{\exp\{-\frac{2}{s}\tilde{L}_{Q_n}(\boldsymbol{W})\}}{Z_s}$, as

$$h_1 \leq \frac{pd}{2}\log\frac{2\pi e(b+spd/2)}{mpd} \tag{18}$$

Moreover, let's define $\tilde{L}^*_{\mathcal{S}_n} \coloneqq \min_{\boldsymbol{W}\in\mathbb{R}^{p\times d}}\tilde{L}_{\mathcal{S}_n}(\boldsymbol{W}) = \tilde{L}_{\mathcal{S}_n}(\boldsymbol{W}^*_{\mathcal{S}_n})$. Then $\nabla\tilde{L}_{\mathcal{S}_n}(\boldsymbol{W}^*_{\mathcal{S}_n}) = 0$, and since $\tilde{L}_{\mathcal{S}_n}$ is $\beta$–smooth, we have $\tilde{L}_{\mathcal{S}_n}(\boldsymbol{W}) - \tilde{L}^*_{\mathcal{S}_n} \leq \frac{\beta}{2}\|\boldsymbol{W}-\boldsymbol{W}^*\|^2$ by Lemma 1.2.3 of Nesterov (2004). As a result, we can lower-bound $\log Z_s$ using a Laplace integral approximation

$$\log Z_s = \log\int_{\mathbb{R}^{p\times d}}\exp\left\{-\frac{2}{s}\tilde{L}_{\mathcal{S}_n}(\boldsymbol{W})\right\}d\boldsymbol{W} = -\frac{2\tilde{L}^*_{\mathcal{S}_n}}{s} + \log\int_{\mathbb{R}^{p\times d}}\exp\left\{\frac{2}{s}(\tilde{L}^*_{\mathcal{S}_n}-\tilde{L}_{\mathcal{S}_n}(\boldsymbol{W}))\right\}d\boldsymbol{W}$$

$$\geq -\frac{2\tilde{L}^*_{\mathcal{S}_n}}{s} + \log\int_{\mathbb{R}^{p\times d}}\exp\left\{-\frac{\beta\|\boldsymbol{W}-\boldsymbol{W}^*_{\mathcal{S}_n}\|^2}{s}\right\}d\boldsymbol{W}$$

$$= -\frac{2\tilde{L}^*_{\mathcal{S}_n}}{s} + \frac{pd}{2}\log\left(\frac{s\pi}{\beta}\right). \tag{19}$$

Using equations (18) and (19) in (17) and simplifying, we obtain

$$\int_{\mathbb{R}^{p\times d}}\tilde{L}_{Q_n}(\boldsymbol{W})\mu_{\mathcal{S}_n}(d\boldsymbol{W}) - \min_{\boldsymbol{W}\in\mathbb{R}^{p\times d}}\tilde{L}_{\mathcal{S}_n}(\boldsymbol{W}) \leq \frac{spd}{4}\log\left(\frac{e\beta}{m}\left(\frac{2b}{spd}+1\right)\right)$$

for $s \leq m$. $\qquad\square$

## B.3 Proof of Risk Minimization on Appropriately Regularized Nets under LMC

*Proof of Theorem 4.1.* The proof will go via 3 steps as follows.

**Step 1: Expected Risk over the law of the iterates approaches the Expected Risk over the Gibbs' measure**

If we choose $Nh$ and $h$, such that

$$Nh = \tilde{\Theta}\left(C_{PI}R_3(\mu_{\mathcal{S}_n,0}\|\mu_{\mathcal{S}_n})\right)$$
$$and \ h = \tilde{\Theta}\left(\frac{\ln(\varepsilon+1)}{pdC_{PI}\ \tilde{\beta}(L_0,\beta)^2\ R_3(\mu_{\mathcal{S}_n,0}\|\mu_{\mathcal{S}_n})} \times \min\left\{1, \frac{1}{2\ln(\varepsilon+1)}, \frac{pd}{m}, \frac{pd}{R_2(\mu_{\mathcal{S}_n,0}\|\mu_{\mathcal{S}_n})^{1/2}}\right\}\right).$$

Now, from Theorem 4.2, replacing $\varepsilon$ by $\ln(\varepsilon+1)$, we get

$$R_2(\pi_{\mathcal{S}_n,N}\|\mu_{\mathcal{S}_n}) \leq \ln(\varepsilon+1). \tag{20}$$

Then, from Lemma B.6 we can say that

$$\mathcal{W}_2(\pi_{\mathcal{S}_n,N},\mu_{\mathcal{S}_n}) \leq 2C_{PI}\varepsilon.$$

Let $\hat{\boldsymbol{W}}$ and $\hat{\boldsymbol{W}}^*$ be random hypotheses, where $\hat{\boldsymbol{W}} \sim \pi_{\mathcal{S}_n,N}$ and $\hat{\boldsymbol{W}}^* \sim \mu_{\mathcal{S}_n} \propto e^{-\tilde{L}_{\mathcal{S}_n}(\boldsymbol{W})}$.

We define $\mathcal{R}(\boldsymbol{W}) \coloneqq \mathbb{E}_{S_n}[\tilde{L}_{\mathcal{S}_n}(\boldsymbol{W})]$ and $\mathcal{R}^* \coloneqq \inf_{\boldsymbol{W}\in\mathbb{R}^{p\times d}}\mathcal{R}(\boldsymbol{W})$.

Then,

$$\mathbb{E}[\mathcal{R}(\hat{\boldsymbol{W}})] - \mathcal{R}^*$$

$$= \underbrace{\mathbb{E}[\mathcal{R}(\hat{\boldsymbol{W}})] - \mathbb{E}[\mathcal{R}(\hat{\boldsymbol{W}}^*)]}_{1} + \underbrace{\mathbb{E}[\mathcal{R}(\hat{\boldsymbol{W}}^*)] - \mathcal{R}^*}_{2}$$

$$= \underbrace{\int_{\mathbb{R}^{p \times d}} \pi_{\mathcal{S}_n, N}(d\boldsymbol{W}) \int_{X^n} \mathcal{R}(\boldsymbol{W}) \mathbb{P}^{\otimes n}(d\mathcal{S}_n) - \int_{\mathbb{R}^{p \times d}} \mu_{\mathcal{S}_n}(d\boldsymbol{W}) \int_{X^n} \mathcal{R}(\boldsymbol{W}) \mathbb{P}^{\otimes n}(d\mathcal{S}_n)}_{1} + \underbrace{\mathbb{E}[\mathcal{R}(\hat{\boldsymbol{W}}^*)] - \mathcal{R}^*}_{2}$$

$$= \underbrace{\int_{X^n} \mathbb{P}^{\otimes n}(d\mathcal{S}_n) \left( \int_{\mathbb{R}^{p \times d}} \mathcal{R}(\boldsymbol{W}) \pi_{\mathcal{S}_n, N}(d\boldsymbol{W}) - \int_{\mathbb{R}^{p \times d}} \mathcal{R}(\boldsymbol{W}) \mu_{\mathcal{S}_n}(d\boldsymbol{W}) \right)}_{1} + \underbrace{\mathbb{E}[\mathcal{R}(\hat{\boldsymbol{W}}^*)] - \mathcal{R}^*}_{2} \qquad (21)$$

where $\mathbb{P}(d\boldsymbol{x})$ is the distribution of a data point, and $\mathbb{P}^{\otimes n}(d\mathcal{S}_n)$ is joint distribution over $n$ data points.

In here we will bound the Term 1 and in Step-2 we will bound Term 2 of above.

The function $F$ satisfies the conditions of Lemma B.1 with $c_1 = \beta$ and $c_2 = B$, and the probability measure $\pi_{\mathcal{S}_n, N}, \mu_{\mathcal{S}_n}$ satisfy the condition of Lemma B.1 with

$$\sigma^2 = \kappa_0 + 2 \cdot \max\left(1, \frac{1}{m}\right)\left(b + 2B^2 + \frac{pds}{2}\right)$$

which is a bound for the second moment of the probability measures that we obtain from Lemma B.2.

Therefore, by replacing $c_1, \sigma, c_2$ and the upperbound to $\mathcal{W}_2(\pi_{\mathcal{S}_n, N}, \mu_{\mathcal{S}_n})$ in equation (16) we get,

$$\int_{\mathbb{R}^{p \times d}} \mathcal{R}(\boldsymbol{W}) \pi_{\mathcal{S}_n, N}(d\boldsymbol{W}) - \int_{\mathbb{R}^{p \times d}} \mathcal{R}(\boldsymbol{W}) \mu_{\mathcal{S}_n}(d\boldsymbol{W}) \le \left( \beta \sqrt{\kappa_0 + 2 \cdot \max\left(1, \frac{1}{m}\right)\left(b + 2B^2 + \frac{pds}{2}\right)} + B \right) 2 C_{PI} \varepsilon \qquad (22)$$

for all $\mathcal{S}_n \in X^n$.

**Step 2 : Expected Population Risk is close to Expected Empirical Risk over the Gibbs' Measure**

Now, it remains to bound the second part of (21). To that end, Raginsky et al. (2017) begins by decomposing it

$$\mathbb{E}[\mathcal{R}(\hat{\boldsymbol{W}}^*)] - \mathcal{R}^* = \underbrace{\mathbb{E}[\mathcal{R}(\hat{\boldsymbol{W}}^*)] - \mathbb{E}[\tilde{L}_{Q_n}(\hat{\boldsymbol{W}}^*)]}_{T_1} + \underbrace{\mathbb{E}[\tilde{L}_{Q_n}(\hat{\boldsymbol{W}}^*)] - \mathcal{R}^*}_{T_2} \qquad (23)$$

where, $Q_n = (\boldsymbol{x}_1, \dots, \boldsymbol{x}_n) \sim \mathbb{P}^{\otimes n}$ is a fixed set of $n$ data points sampled from the joint distribution $\mathbb{P}^{\otimes n}$.

In this step we would bound $T_1$. $T_2$ would be bounded in the next step 3.

To bound $T_1$ in (23), let's sample[1] $\mathcal{S}'_n = (\boldsymbol{x}'_1, \dots, \boldsymbol{x}'_n) \sim \mathbb{P}^{\otimes n}$ independent of $\mathcal{Q}_n$ and $\hat{\boldsymbol{W}}^*$. Then we have,

$$\mathbb{E}_{\mathcal{S}_n, \hat{\boldsymbol{W}}^* \sim \mu_{\mathcal{S}_n}}[\mathcal{R}(\hat{\boldsymbol{W}}^*)] - \mathbb{E}_{\mathcal{S}_n, \hat{\boldsymbol{W}}^* \sim \mu_{\mathcal{S}_n}}[\tilde{L}_{Q_n}(\hat{\boldsymbol{W}}^*)]$$

$$= \mathbb{E}_{\substack{\mathcal{S}'_n \\ \mathcal{S}_n, \hat{\boldsymbol{W}}^* \sim \mu_{\mathcal{S}_n}}}[\tilde{L}_{\mathcal{S}'_n}(\hat{\boldsymbol{W}}^*) - \tilde{L}_{Q_n}(\hat{\boldsymbol{W}}^*)] \qquad (24)$$

$$= \frac{1}{n} \sum_{i=1}^{n} \mathbb{E}_{\substack{\boldsymbol{x}'_i \sim \mathbb{P} \\ \mathcal{S}_n, \hat{\boldsymbol{W}}^* \sim \mu_{\mathcal{S}_n}}}[\tilde{L}'_i(\hat{\boldsymbol{W}}^*) - \tilde{L}_i(\hat{\boldsymbol{W}}^*)] \qquad (25)$$

---

[1]Here we will be abusing the notation of random variables and their instances slightly

The $i$-th term in the above summation can be written out explicitly as,

$$
\mathbb{E}_{\substack{\boldsymbol{x}'_i \sim \mathbb{P} \\ \mathcal{S}_n, \hat{\boldsymbol{W}}^* \sim \mu_{\mathcal{S}_n}}} [\tilde{L}'_i(\hat{\boldsymbol{W}}^*) - \tilde{L}_i(\hat{\boldsymbol{W}}^*)]
$$

$$
= \int_{X^n} \mathbb{P}^{\otimes n}(d\mathcal{S}_n) \int_X \mathbb{P}(d\boldsymbol{x}'_i) \int_{\mathbb{R}^{p \times d}} \mu_{\mathcal{S}_n}(d\boldsymbol{W})[\tilde{L}'_i(\boldsymbol{W}) - \tilde{L}_i(\boldsymbol{W})]
$$

$$
= \int_{X^n} \mathbb{P}^{\otimes n}(d\boldsymbol{x}_1, \cdots, d\boldsymbol{x}_i, \cdots, d\boldsymbol{x}_n) \int_X \mathbb{P}(d\boldsymbol{x}'_i) \int_{\mathbb{R}^{p \times d}} \pi_{(\boldsymbol{x}_1, \cdots, \boldsymbol{x}_i, \cdots, \boldsymbol{x}_n)}(d\boldsymbol{W}) \tilde{L}'_i(\boldsymbol{W})
$$

$$
- \int_{X^n} \mathbb{P}^{\otimes n}(d\boldsymbol{x}_1, \cdots, d\boldsymbol{x}_i, \cdots, d\boldsymbol{x}_n) \int_X \mathbb{P}(d\boldsymbol{x}'_i) \int_{\mathbb{R}^{p \times d}} \pi_{(\boldsymbol{x}_1, \cdots, \boldsymbol{x}_i, \cdots, \boldsymbol{x}_n)}(d\boldsymbol{W}) \tilde{L}_i(\boldsymbol{W}) \tag{26}
$$

Since all $\boldsymbol{x}_i$-s are sampled independently of each other we can interchange $\boldsymbol{x}_i$ and $\boldsymbol{x}'_i$ in the first term, and the

$$
= \int_{X^n} \mathbb{P}^{\otimes n}(d\boldsymbol{x}_1, \cdots, d\boldsymbol{x}'_i, \cdots, d\boldsymbol{x}_n) \int_X \mathbb{P}(d\boldsymbol{x}_i) \int_{\mathbb{R}^{p \times d}} \pi_{(\boldsymbol{x}_1, \cdots, \boldsymbol{x}'_i, \cdots, \boldsymbol{x}_n)}(d\boldsymbol{W}) \tilde{L}_i(\boldsymbol{W})
$$

$$
- \int_{X^n} \mathbb{P}^{\otimes n}(d\boldsymbol{x}_1, \cdots, d\boldsymbol{x}_i, \cdots, d\boldsymbol{x}_n) \int_X \mathbb{P}(d\boldsymbol{x}'_i) \int_{\mathbb{R}^{p \times d}} \pi_{(\boldsymbol{x}_1, \cdots, \boldsymbol{x}_i, \cdots, \boldsymbol{x}_n)}(d\boldsymbol{W}) \tilde{L}_i(\boldsymbol{W})
$$

$$
= \int_{X^n} \mathbb{P}^{\otimes n}(d\mathcal{S}_n) \int_X \mathbb{P}(d\boldsymbol{x}'_i) \left( \int_{\mathbb{R}^{p \times d}} \pi_{\mathcal{S}_n^{(i)}}(d\boldsymbol{W}) \tilde{L}_i(\boldsymbol{W}) - \int_{\mathbb{R}^{p \times d}} \mu_{\mathcal{S}_n}(d\boldsymbol{W}) \tilde{L}_i(\boldsymbol{W}) \right) \tag{27}
$$

where $\mathcal{S}_n^{(i)}$ and $\mathcal{S}_n$ differ only in the $i$−th coordinate. Then from Proposition B.5 we obtain

$$
\mathbb{E}[\mathcal{R}(\hat{\boldsymbol{W}}^*)] - \mathbb{E}[\tilde{L}_{Q_n}(\hat{\boldsymbol{W}}^*)] \leq \frac{1}{n} \sum_{i=1}^{n} \frac{\tilde{C}_3}{n} = \frac{\tilde{C}_3}{n}. \tag{28}
$$

**Step 3 : Empirical Risk Minimization under LMC**

Now, to bound the second term $T_2$, we choose a minimizer $\boldsymbol{W}^* \in \mathbb{R}^{p \times d}$ of $\mathcal{R}(\boldsymbol{W})$, i.e. $\mathcal{R}(\boldsymbol{W}^*) = \mathcal{R}^*$. Then

$$
\mathbb{E}[\tilde{L}_{Q_n}(\hat{\boldsymbol{W}}^*)] - \mathcal{R}^*
$$

$$
= \mathbb{E}_{\mathcal{S}_n, \hat{\boldsymbol{W}}^* \sim \mu_{\mathcal{S}_n}}[\tilde{L}_{Q_n}(\hat{\boldsymbol{W}}^*)] - \mathbb{E}_{\mathcal{S}_n}[\min_{\boldsymbol{W} \in \mathbb{R}^{p \times d}} \tilde{L}_{\mathcal{S}_n}(\boldsymbol{W})] + \mathbb{E}_{\mathcal{S}_n}[\min_{\boldsymbol{W} \in \mathbb{R}^{p \times d}} \tilde{L}_{\mathcal{S}_n}(\boldsymbol{W})] - \mathcal{R}(\boldsymbol{W}^*) \tag{29}
$$

$$
= \mathbb{E}_{\mathcal{S}_n, \hat{\boldsymbol{W}}^* \sim \mu_{\mathcal{S}_n}}[\tilde{L}_{Q_n}(\hat{\boldsymbol{W}}^*) - \min_{\boldsymbol{W} \in \mathbb{R}^{p \times d}} \tilde{L}_{\mathcal{S}_n}(\boldsymbol{W})] + \mathbb{E}_{\mathcal{S}_n}[\min_{\boldsymbol{W} \in \mathbb{R}^{p \times d}} \tilde{L}_{\mathcal{S}_n}(\boldsymbol{W}) - \tilde{L}_{\mathcal{S}_n}(\boldsymbol{W}^*)] \tag{30}
$$

Since $\mathbb{E}_{\mathcal{S}_n}[\min_{\boldsymbol{W} \in \mathbb{R}^{p \times d}} \tilde{L}_{\mathcal{S}_n}(\boldsymbol{W}) - \tilde{L}_{\mathcal{S}_n}(\boldsymbol{W}^*)] \leq 0$, we can say that

$$
\mathbb{E}[\tilde{L}_{Q_n}(\hat{\boldsymbol{W}}^*)] - \mathcal{R}^* \leq \mathbb{E}_{\mathcal{S}_n, \hat{\boldsymbol{W}}^* \sim \mu_{\mathcal{S}_n}}[\tilde{L}_{Q_n}(\hat{\boldsymbol{W}}^*) - \min_{\boldsymbol{W} \in \mathbb{R}^{p \times d}} \tilde{L}_{\mathcal{S}_n}(\boldsymbol{W})] \tag{31}
$$

$$
= \mathbb{E}_{\mathcal{S}_n}[\int_{\mathbb{R}^{p \times d}} \tilde{L}_{Q_n}(\boldsymbol{W}) \mu_{\mathcal{S}_n}(d\boldsymbol{W}) - \min_{\boldsymbol{W} \in \mathbb{R}^{p \times d}} \tilde{L}_{\mathcal{S}_n}(\boldsymbol{W})] \tag{32}
$$

$$
\leq \frac{pds}{4} \log\left( \frac{e\beta}{m} \left( \frac{2b}{spd} + 1 \right) \right) \tag{33}
$$

where the last step is by Proposition B.7.

Then combining (22), (28) and (33) we get,

$$
\mathbb{E}[\mathcal{R}(\boldsymbol{W}_N)] - \mathcal{R}^* \leq \frac{\tilde{C}_3}{n} + \frac{pds}{4} \log\left( \frac{e\beta}{m} \left( \frac{2b}{spd} + 1 \right) \right)
$$

$$
+ \left( \beta \sqrt{\kappa_0 + 2 \cdot \max\left( 1, \frac{1}{m} \right) \left( b + 2B^2 + \frac{pds}{2} \right)} + B \right) 2 C_{PI} \varepsilon \tag{34}
$$

**Remark.** The second term on the RHS of (34) can be made $\tilde{O}(\varepsilon)$ by setting $s = \tilde{O}(\varepsilon)$, since for $\varepsilon \geq \frac{1}{e^k - 1}$ with some $k > 0$, we have $\varepsilon \ln(1 + 1/\varepsilon) \leq k\varepsilon$. This was further seen in our experiments as mentioned in Appendix 5.

Once $s$ is fixed, considering the first term, i.e., $\frac{\tilde{C}_3}{n}$ where $\tilde{C}_3 \coloneqq 16\sqrt{2}\left(\beta^2 \frac{b+spd/2}{m} + B^2\right)\frac{C_{PI}\sqrt{\mathcal{C}(\tilde{L}_{\mathcal{S}_n})}}{s}$, we observe that for large enough $n$, $\mathcal{C}(\tilde{L}_{\mathcal{S}_n}) = \sup_{\boldsymbol{W} \in \mathbb{R}^{p \times d}} \frac{\exp\left(\frac{-2}{ns}(\tilde{L}_i(\boldsymbol{W}) - \tilde{L}_i'(\boldsymbol{W}))\right)}{K} = \tilde{O}(e^{1/n})$. It follows that $\frac{\tilde{C}_3}{n} = \tilde{O}(\frac{e^{1/n}}{n})$, and setting $n = \tilde{O}(\frac{1}{\varepsilon})$ ensures that the first term becomes $\tilde{O}(\varepsilon)$.

The LMC converges at a rate of $\tilde{O}\left(\frac{p^3 d^3 C_{PI}^2 \tilde{\beta}(L_0, \beta)^2}{\ln(1+\varepsilon)} \times \max\left(1, \frac{r}{pd}\right)\right)$, which is derived from the convergence rate in Theorem 4.2 by setting $q = 2$ and substituting $\varepsilon$ with $\ln(\varepsilon + 1)$.

$\square$

