# OpenReview forum: "Langevin Monte-Carlo Provably Learns Depth Two Neural Nets at Any Size and Data"
_TMLR — Under review for TMLR_

### Review · Reviewer_TKbW · 2026-07-11

**Summary Of Contributions:**

This paper analyzes Langevin Monte Carlo for training regularized finite-width two-layer neural networks with fixed output weights. It proves convergence of the iterate distribution to the corresponding Gibbs measure in Rényi divergence under suitable regularization. The authors then derive an excess regularized population-risk bound from this sampling guarantee. The analysis covers smooth bounded activations in regression and binary classification. Experiments on a simple regression task illustrate similar behavior across widths and comparable performance to AdamW.

**Audience:**

Yes

**Audience Explanation:**

The analysis of simple neural networks has long been the subject of widespread attention. Therefore, the TMLR's audience will be interested in knowing the findings of this paper.

**Claims And Evidence:**

Yes

**Claims Explanation:**

The main theoretical claims are supported by a clear sequence of arguments connecting regularized neural-network losses, Poincaré inequalities, and existing convergence results for Langevin Monte Carlo. The resulting Rényi-divergence and excess-risk bounds are stated explicitly, with their assumptions and parameter dependencies identified. The experiments are limited in scope, but they provide a useful qualitative illustration of the predicted behavior across different network widths.

**Requested Changes:**

1. The authors should clarify the scope of the main claims, especially the meanings of “any size” and “any data.” The current analysis assumes bounded data, smooth bounded activations, fixed output-layer weights, and sufficiently strong regularization. These conditions should be stated more prominently in the title, abstract, and introduction.

2. The paper should provide a clearer discussion of the dependence on the Poincaré constant and other implicit quantities in the convergence and excess-risk bounds. In particular, it would be helpful to explain whether these constants scale with the network width and parameter dimension, and how this affects the practical interpretation of the results.

3. The notation should be made more consistent throughout the paper, particularly for the temperature parameter, the Gibbs measure, and the different loss functions. A short table summarizing the main symbols and assumptions would improve readability.

---

### Review · Reviewer_Q8Zs · 2026-07-14

**Summary Of Contributions:**

The paper studies Langevin Monte Carlo (LMC) for training finite-width depth-2 neural networks with fixed second-layer weights, bounded smooth activations, and Frobenius-norm regularization. The central proof strategy is modular:

1. Use prior results to show that, once the regularization parameter exceeds an explicit threshold, the regularized neural-network loss is a Villani function and its Gibbs measure satisfies a Poincaré inequality.
2. Invoke existing non-asymptotic LMC convergence theory to obtain convergence of the last-iterate distribution to the Gibbs measure in q-Rényi divergence.
3. Combine this distributional convergence with a Gibbs-stability argument and an approximate empirical-risk-minimization property of the Gibbs distribution to derive an excess regularized population-risk guarantee.

The paper is motivated by the fact that much of the existing theory for neural-network optimization relies on overparameterized, neural-tangent-kernel, mean-field, teacher–student, or other restrictive regimes. In contrast, the present result does not require the hidden width to exceed a threshold depending on sample size or target accuracy. The paper also emphasizes that the critical regularization thresholds have no explicit width dependence when the norm of the fixed second-layer weights is controlled.

The overall synthesis is interesting and relevant to readers working on Langevin methods, functional inequalities, Gibbs algorithms, and provable neural-network training. However, the current manuscript contains a central mathematical error in the Rényi-to-Wasserstein step used to establish Theorem 4.1, together with several additional inconsistencies and proof gaps. These issues affect the stated non-asymptotic rate and prevent the main excess-risk theorem from being considered established as written.

**Audience:**

Yes

**Audience Explanation:**

The paper addresses topics central to the TMLR audience, including nonconvex learning dynamics, Langevin algorithms, Gibbs distributions, functional inequalities, stability-based generalization, and finite-width neural-network theory.

The attempted synthesis is conceptually appealing:

`regularized neural loss → Villani condition → Poincaré inequality → LMC distributional convergence → population-risk control`

In particular, a last-iterate distributional convergence result for a stochastic training algorithm on finite-width neural networks would be of interest to part of the TMLR audience, even though the current manuscript does not yet establish the advertised quantitative learning guarantee correctly.

**Claims And Evidence:**

No

**Claims Explanation:**

## Explanation

The high-level proof architecture is plausible, and the qualitative convergence argument may be repairable. However, the manuscript does not currently provide a correct and internally complete proof of its principal excess-risk guarantee.

### 1. Lemma B.6 appears to miss a square root

The most consequential issue is Lemma B.6. The manuscript states

$$W_2(\\pi,\\mu) \\leq 2C_{\\mathrm{PI}}\\left(e^{R_2(\\pi\\,\\|\\,\\mu)}-1\\right).$$

The Poincaré-based transportation-variance inequality used in the proof controls $W_2^2$, not $W_2$ directly. From

$$W_2^2(\\pi,\\mu) \\leq 2C_{\\mathrm{PI}}\\operatorname{Var}_{\\mu}\\left(\\frac{d\\pi}{d\\mu}\\right)$$

and

$$\\operatorname{Var}_{\\mu}\\left(\\frac{d\\pi}{d\\mu}\\right)=e^{R_2(\\pi\\,\\|\\,\\mu)}-1,$$

one obtains

$$W_2(\\pi,\\mu) \\leq \\sqrt{2C_{\\mathrm{PI}}\\left(e^{R_2(\\pi\\,\\|\\,\\mu)}-1\\right)}.$$

This difference directly affects Section 4.1 and Appendix B.3. The paper currently sets

$$R_2\\left(\\pi_{S_n,N}\\,\\|\\,\\mu_{S_n}\\right) \\leq \\log(1+\\varepsilon)$$

and concludes that $W_2=O(\\varepsilon)$. With the corrected inequality, the same Rényi-divergence bound only gives $W_2=O(\\sqrt{\\varepsilon})$.

The qualitative argument may be salvageable by requiring a Rényi-divergence error of order $\\varepsilon^2$, but this would alter the step-size requirement, iteration complexity, and final accuracy dependence. Consequently, Equations (22) and (34), Theorem 4.1, and the claimed $\\widetilde{O}(1/\\varepsilon)$-type rate are not established as written.

### 2. Theorem 4.2 and Appendix A.1 are not written consistently

Theorem 4.2 states

$$T=\\widetilde{\\Theta}\\left(qC_{\\mathrm{PI}}R_{2q-1}\\left(\\pi_0\\,\\|\\,\\mu_s\\right)\\right).$$

Appendix A.1 instead introduces

$$T=\\widetilde{\\Theta}\\left(qC_{\\mathrm{PI}}R_{2q-1}\\left(\\pi_0\\,\\|\\,\\mu_s\\right)^{2/\\alpha-1}\\right),$$

where $\\alpha$ is not defined in the manuscript.

This appears likely to be an incomplete specialization of a more general Latała--Oleszkiewicz inequality result, for which $\\alpha=1$ corresponds to the Poincaré case. Under $\\alpha=1$, the exponent satisfies $2/\\alpha-1=1$, and the appendix formula agrees with the theorem statement.

Thus, this issue may be a correctable specialization or notation error rather than a fundamental contradiction. Nevertheless, the manuscript must explicitly state the invoked theorem, define $\\alpha$, specialize it to $\\alpha=1$, and verify all resulting exponents and complexity expressions.

### 3. The manuscript conflates continuous-time horizon and discrete iteration complexity

The continuous-time horizon in Theorem 4.2 is $T=N_qh_q$.

The target-accuracy dependence primarily enters through the permitted step size $h_q$, and therefore through the discrete iteration count $N_q=T/h_q$.

The discussion in Section 1.1 and after Theorem 4.1 sometimes presents the $\\widetilde{O}(1/\\varepsilon)$ dependence as though it were a generic convergence-time statement. The paper should distinguish clearly among:

- the continuous-time horizon $T$;
- the numerical discretization step size $h_q$;
- the number of LMC iterations or gradient evaluations $N_q$.

This distinction becomes even more important after correcting Lemma B.6, since obtaining an $O(\\varepsilon)$ risk contribution may require Rényi-divergence accuracy of order $\\varepsilon^2$, thereby worsening the discrete step complexity.

### 4. The width-independence claim requires qualification

The regularization thresholds are

$$\\lambda_c^{\\mathrm{MSE}}=2M_DLB_x^2\\lVert a\\rVert_2^2$$

and

$$\\lambda_c^{\\mathrm{BCE}}=\\frac{1}{2}M_DLB_x^2\\lVert a\\rVert_2^2.$$

These expressions contain no explicit factor of the width $p$. However, the fixed second-layer vector satisfies $a\\in\\mathbb{R}^p$, and $\\lVert a\\rVert_2$ may itself grow with width. If the coordinates of $a$ remain $O(1)$, then $\\lVert a\\rVert_2^2=O(p)$.

The correct claim is therefore that the critical threshold has no explicit width dependence when the second-layer norm is held uniformly bounded across widths. This is consistent with the experiments, where $\\lVert a\\rVert_2$ is fixed.

Moreover, the full convergence bounds are not width-uniform. They depend on $pd$, smoothness quantities, initialization divergences, and the Poincaré constant $C_{\\mathrm{PI}}$. The paper proves the existence of a Poincaré constant but does not bound it, and Section 6 acknowledges that controlling this constant remains open. Thus, the paper does not establish polynomial-time or width-uniform learning.

### 5. Theorem 4.1 controls a regularized population objective

The paper defines

$$R(W)=\\mathbb{E}_{S_n}\\left[\\widetilde{L}_{S_n}(W)\\right],$$

where $\\widetilde{L}_{S_n}$ includes the term

$$\\frac{\\lambda}{2}\\lVert W\\rVert_F^2.$$

Accordingly, Theorem 4.1 bounds excess regularized population risk, not excess prediction risk alone. The current "learns neural nets" language may lead readers to interpret the result as a guarantee on unregularized prediction error or Bayes excess risk, which is not what is proved.

This distinction should be stated prominently in the abstract, introduction, and theorem discussion.

### 6. The Gibbs-stability proof does not fully control the normalization ratio

In Lemmas B.3 and B.4, the manuscript introduces

$$C\\left(\\widetilde{L}_{S_n}\\right)=\\sup_W\\frac{d\\mu_{S_n}}{d\\mu_{\\overline{S}_n}}(W),$$

which contains the partition-function ratio

$$K=\\frac{Z_{S_n}}{Z_{\\overline{S}_n}}.$$

The bound in Lemma B.4 retains a factor $1/K$, but the final discussion later treats

$$C\\left(\\widetilde{L}_{S_n}\\right)=\\widetilde{O}\\left(e^{1/n}\\right)$$

without explicitly controlling $K$.

This gap may be repairable because the loss difference between neighboring datasets is uniformly bounded in the stated setting, which should also yield corresponding upper and lower bounds on the partition-function ratio. However, that argument must be written explicitly. Since $C\\left(\\widetilde{L}_{S_n}\\right)$ enters the stability and generalization constants, the omission is not merely presentational.

### 7. Some auxiliary BCE bounds require correction

In Claim 1 of Appendix B.1, the BCE loss at $W=0$ is bounded using an expression that does not appear valid uniformly over $y_i\\in\\{-1,+1\\}$ and arbitrary signs of $\\langle a,c\\rangle$. A safe bound would involve

$$\\log\\left(1+e^{\\lvert\\langle a,c\\rangle\\rvert}\\right).$$

The associated gradient bound should similarly avoid a sign-dependent denominator unless the relevant sign conditions are stated.

These appear to be repairable constant-level errors, but they reinforce the need for a systematic audit of the appendix.

### 8. The experiments are illustrative but do not validate the main theorem

Figures 1--5 show that training and test losses decrease across several widths in a one-dimensional synthetic regression task, that label noise changes the attained loss, and that AdamW exhibits qualitatively similar behavior in the same setup.

These experiments are useful illustrations, but they provide limited evidence for the theoretical claims:

- there is no binary-classification experiment despite the BCE theory;
- no multiple-run statistics or uncertainty estimates are reported;
- no quantitative result table is provided;
- the experiments do not estimate Rényi divergence, Wasserstein distance, Gibbs mixing, or $C_{\\mathrm{PI}}$;
- the predicted accuracy or step-complexity dependence is not tested;
- similarity between AdamW and LMC curves does not imply that the LMC theory applies to AdamW.

For a primarily theoretical paper, these limitations are not independently decisive. However, the experiments do not compensate for the proof issues.

**Requested Changes:**

## Requested Changes

1. **Critical: Correct Lemma B.6 and all downstream uses.**

   Restate the correct Poincaré-to-Wasserstein inequality, including the square root. Revise Section 4.1, Appendix B.3, Equations (22) and (34), and Theorem 4.1 accordingly. Recompute the target Rényi accuracy required for an `O(epsilon)` risk contribution and update the resulting step-size and iteration-complexity dependence.

2. **Critical: Re-establish the quantitative statement of Theorem 4.1.**

   After correcting Lemma B.6, provide a complete theorem statement with the revised accuracy dependence. If the result requires running Theorem 4.2 to Rényi error `O(epsilon^2)`, this should be reflected explicitly in the theorem and complexity discussion.

3. **Major: Resolve the undefined alpha in Appendix A.1.**

   State the precise external theorem being invoked, define the LOI parameter `alpha`, specialize it explicitly to the Poincaré case `alpha = 1`, and ensure that Theorem 4.2, Appendix A.1, and the formula for `N_q` use consistent exponents.

4. **Major: Distinguish continuous-time and discrete complexity.**

   Clearly separate the continuous-time horizon `T`, the step size `h_q`, and the iteration count `N_q`. Attach any `1/epsilon` or `1/epsilon^2` dependence only to the quantity supported by the corrected derivation.

5. **Major: Narrow the width-independence claims.**

   State that the critical regularization threshold has no explicit width dependence only when the second-layer norm `||a||_2` remains uniformly controlled across widths. Do not imply width-uniform convergence constants, since the full bound depends on dimension and on an unquantified Poincaré constant `C_PI`.

6. **Major: Clarify the learning objective.**

   Make clear throughout that Theorem 4.1 controls excess regularized population risk rather than unregularized prediction risk or Bayes excess risk.

7. **Major: Complete the Gibbs-stability argument.**

   Provide an explicit bound on the partition-function ratio `Z_{S_n} / Z_{Sbar_n}`, justify the claimed bound on `C(L_tilde_{S_n})`, and propagate the resulting constants through Proposition B.5 and Theorem 4.1.

8. **Major: Audit and correct the BCE auxiliary bounds.**

   Correct the sign-dependent bounds in Claim 1 and verify all smoothness, bounded-gradient, and dissipativity constants used later in the proof.

9. **Strengthen: Clarify novelty relative to imported results.**

   The paper should identify more crisply which results are new in the manuscript. Lemmas 3.1 and 3.2 are imported from prior work, and Theorem 4.2 is largely an application of existing LMC convergence theory once the Poincaré inequality and smoothness conditions have been established. The new contribution appears primarily to be the specialization, synthesis, and risk-minimization argument.

10. **Strengthen: Improve the empirical section.**

    Add at least one binary-classification experiment, report quantitative summaries across random seeds, and explain the precise parameterization used for `s`, `h`, and the gradient multiplier. An empirical study of the predicted accuracy or discretization dependence would also improve the paper, although direct estimation of the theoretical constants may be difficult.

---

### Review · Reviewer_bvay · 2026-07-20

**Summary Of Contributions:**

This paper studies Langevin Monte Carlo (LMC) for training two-layer NNs, specifically NNs with fixed last layer weights and trainable first layer weights. The loss function they consider are squared loss and logistic loss with Frobenius-norm regularization. The analysis assumes bounded data and bounded, smooth activation functions (ex. tanh or sigmoid). Their main theorem requires showing that the Gibbs measure of these loss functions satisfy a Poincare-type inequality. This follows from previous rresults showing that for sufficiently high regularized Frobenius norm that these loss functions will become Villani functions. It is then known from prior work that for Villani functions their Gibbs measure satisfies a Poincare-type inequality as desired.

The point of establishing these properties is that the authors can then apply existing LMC convergence results for smooth potentials whose Gibbs measures satisfy a Poincare inequality. They show the distribution of the last LMC iterate has non-asymptotic convergence in q-Renyi divergence to the Gibbs distribution of the regularized empirical loss. This holds for any finite network width.

Next they convert convergence in Renyi divergence to convergence in Wasserstein distance so that they can compare the expected population risk under the LMC iterate distribution to the Gibbs distribution. They also prove a stability result for the Gibbs distribution. Then they use the fact that sampling from the Gibbs distribution approximately minimizes the empirical loss. Combining all these parts allows them to state their final population-risk bound for LMC.


So their main claimed contribution is a learning guarnatee for LMC that applies to finite-width two-layer NNs, which is a looser requirement than infinite-width, mean-field,NTK, etc. The paper also includes synthetic regression experiments showing LMC training networks across a range of widths to support their theory.


Key strengths:

I found the overall connection between NN training and the recent work in sampling theory to be interesting. It is good that the authors could obtain finite-width results. This is distinctive from a lot of two-layer work that has to use infinite width or some other stronger assumptions, and also only needed bounded data assumption nothing stronger there either. Also it's great they can get last-iterate convergence results not just time averaged. The paper has a pretty well-explained and clear proof strategy and is relatively easy to follow. The proof in Appendix B3 "Stability of Gibbs’ Algorithm under PI" appears technically non-trivial.

Key weaknesses:

Overall it does feel like the technical heavy lifting for the results in this paper comes primarily from the main papers they cite and build on, though of course seeing these connections and putting things together for the results they obtain is nontrivial. Also the runtime bound feels quite large/loose/uninterpretable in that there is a polynomial dependence on the number of trainable parameters of the NN and more critically a quadratic dependence on the Poincare constant which could be large. The authors do acknowledge this issue about the Poincare constant, though. I also feel the framing of the contribution is a bit more broad than warranted and not needed to sell the main contributions of the work. For example they require smooth activation functions so ReLU is not covered and the population risk itself being analyzed includes the Frobenius regularization term, so the guarantee is actually on this regularized risk from my read, so not like unregularized risk. Although it's standard to have assumptions like this I wonder about the opening to abstract and some claims in Section 1.1 Of course once the actual math starts later in the paper everything is made clear.

**Audience:**

Yes

**Audience Explanation:**

This is clear, it's an interesting and relevant result, quoting the authors directly "there has never been a convergence result for the law of
the iterates of any stochastic training algorithm for neural nets." This is also true to my knowledge.

**Broader Impact Concerns:**

None.

**Claims And Evidence:**

Yes

**Claims Explanation:**

I put yes because it is mainly true, though I would still call attention to the weaknesses stated above. I think the authors could do a slightly better job wording the claims in the opening to the paper.

**Requested Changes:**

I think overall the paper is in a good shape.

Perhaps again just some slightly more tailored wordings to the abstract and intro.

I think there may also be an issue with the statement of the result from Liu 2020 in the Lemma B.6 proof, it is inconsistent with the statement of the same result in Lemma B.3 and from checking the Liu paper it seems the B.3 is the correct version? Maybe I am missing something here but it would be great if the authors could doublecheck.

---

### Review · Reviewer_mbBu · 2026-07-20

**Summary Of Contributions:**

In this paper, authors claims that the Langevin Monte Carlo algorithm can provably learn depth-2 neural networks of arbitrary width and for arbitrary data, with non-asymptotic convergence rates. The key argument proceeds by showing that Frobenius-norm-regularized empirical losses (MSE and BCE) for two-layer nets with smooth activations (tanh, sigmoid) are Villani functions, which implies their Gibbs measures satisfy a Poincaré inequality. Leverage existing results, authors prove that it is equal to population risk minimization.  Author practically validate the results through synthetic dataset using depth-2 tanh networks. with different width.

**Audience:**

Yes

**Audience Explanation:**

This is a very theoretical work with detailed proof and analysis on vanilla neural network, which can have guidance to modern deep learning. Thus, I think some TMLR's audience would be interested in the results.

**Broader Impact Concerns:**

None.

**Claims And Evidence:**

No

**Claims Explanation:**

One important issue is the so claimed any data size. from the theoretical perspective, while no structural While no structural assumption on data is needed, the convergence bound depends on data-dependent quantities. From empirical perspective, author only validate the theory in the tiny synthetic dataset, which is far away from real-world scenarios.

**Requested Changes:**

1. Authors should make clear explanation on the claimed "any data" and "any size" for the proposed theory. Are these claims indicate that the conclusion in the paper can be applied to any real-world data or neural networks?

2. Authors should provide additional experiments on real-world datasets to should how the proposed theorem align with real-world scenarios, at least using MNIST, CIFAR10, etc.

3. It would be better for authors to provide additional detailed discussion to explain how the proposed theorem can connect to the modern deep learning, like larger NN with non-smooth activation, or transformer, large language model, etc.